# SpineBench: A Clinically Salient, Level-Aware Benchmark Powered by the SpineMed-450k Corpus

**Ming Zhao♣, Wenhui Dong♣, Yang Zhang♣, Xiang Zheng, Zhonghao Zhang, Zian Zhou, Yunzhi Guan, Liukun Xu, Wei Peng, Zhaoyang Gong, Zhicheng Zhang, Dachuan Li, Xiaosheng Ma, Yuli Ma, Jianing Ni, Changjiang Jiang, Lixia Tian, Qixin Chen, Kaishun Xia, Pingping Liu, Tongshun Zhang, Zhiqiang Liu, Zhongyan Bi, Chenyang Si, Tiansheng Sun✉, and Caifeng Shan✉**

## Abstract

Spine disorders affect 619 million people globally and are a leading cause of disability, yet AI-assisted diagnosis remains limited by the lack of level-aware, multimodal datasets. Clinical decision-making for spine disorders requires sophisticated reasoning across X-ray, CT, and MRI at specific vertebral levels. However, progress has been constrained by the absence of traceable, clinically-grounded instruction data and standardized, spine-specific benchmarks. To address this, we introduce SpineMed, an ecosystem co-designed with practicing spine surgeons. It features SpineMed-450k, the first large-scale dataset explicitly designed for vertebral-level reasoning across imaging modalities with over 450,000 instruction instances, and SpineBench, a clinically-grounded evaluation framework. SpineMed-450k is curated from diverse sources, including textbooks, guidelines, open datasets, and ~1,000 de-identified hospital cases, using a clinician-in-the-loop pipeline with a two-stage LLM generation method (draft and revision) to ensure high-quality, traceable data for question-answering, multi-turn consultations, and report generation. SpineBench evaluates models on clinically salient axes, including level identification, pathology assessment, and surgical planning. Our comprehensive evaluation of several recently advanced large vision-language models (LVLMs) on SpineBench reveals systematic weaknesses in fine-grained, level-specific reasoning. In contrast, our model fine-tuned on SpineMed-450k demonstrates consistent and significant improvements across all tasks. Clinician assessments confirm the diagnostic clarity and practical utility of our model's outputs.

# 1 Introduction

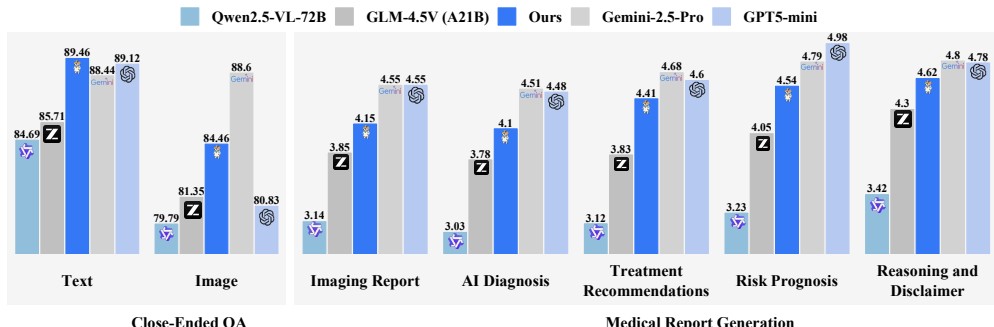

Figure 1: Benchmark performance of SpineGPT

♣ Equal Contributions. ✉Corresponding author: suntiansheng-@163.com; cfshan@nju.edu.cn. The complete author details are provided in Appendix B.

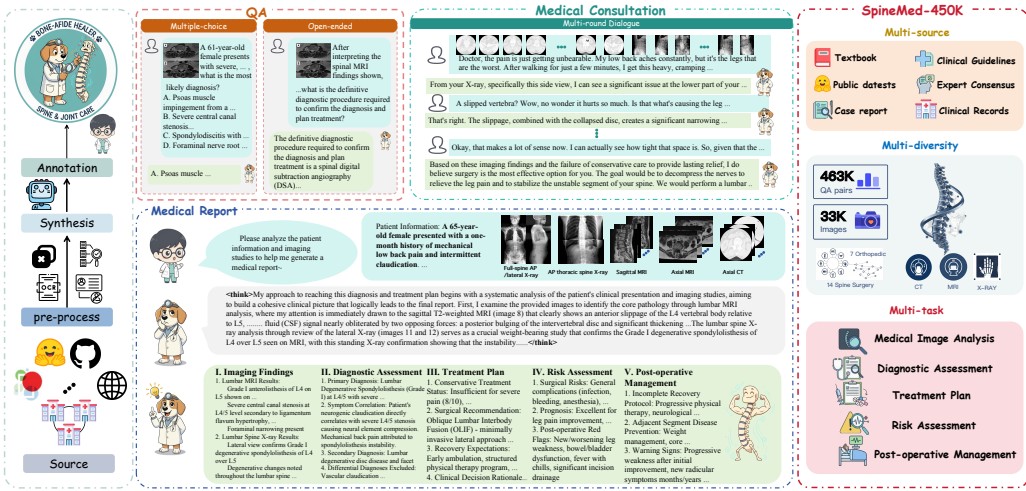

Figure 2: Overview of SpineMed-450k. Training data was curated from textbooks, public datasets, clinical records, medical guidelines, and hospitals. The process involved data preprocessing, annotation generation, and a final clinician review. Our dataset comprises four types: multi-choice QA, open-ended QA, multi-round dialogues, and reports.

Spinal disorders (Ferreira et al., 2023), including degenerative diseases (like disc herniation) (Dydyk et al., 2017), deformities (like scoliosis) (Negrini et al., 2018), trauma (fractures) (Vaccaro et al., 2013), and inflammatory conditions (Taurog et al., 2016), are a major driver of pain, disability, and surgical care worldwide. A key challenge in their management is diagnostic complexity. Unlike many other disorders, spinal conditions typically cannot be precisely diagnosed using a single imaging modality. It often requires clinicians to perform level-aware, multimodal reasoning: integrating findings from X-ray, CT, and MRI to pinpoint pathology at specific vertebral levels, grade severity, and plan interventions (Teichner et al., 2025). The precision of this interpretation directly impacts patient outcomes and neurological safety. Although advanced AI holds great promise for augmenting this demanding workflow (Ibrahim et al., 2025b), its potential has been hindered. Fortunately, such clinical tasks can significantly benefit from advanced AI capabilities (Lee et al., 2024b). Yet progress is constrained not by model capacity, but by the absence of *traceable instruction data* and *standardized, clinically validated benchmarks* tailored to spine workflows (Lee et al., 2024b). Equally important, prior efforts rarely embed clinicians throughout the pipeline, limiting practical utility. We present **SpineMed**: a comprehensive effort consisting of **SpineMed-450k**, a provenance-rich instruction corpus for spine diagnosis and planning, and **SpineBench**, a targeted evaluation suite that help to evaluate the effectiveness of different AI-based spine diagnosis. To our best knowledge, this is current largest-scale Spinal diagnosis and treatment dataset. Both were *co-designed with spine clinicians* (radiologists and surgeons) to reflect real decision points. SpineMed-450k aggregates materials from textbooks, surgical guidelines, expert consensuses, question banks, open spine datasets (e.g., Spark, VerSe) (Alibaba Cloud Tianchi, 2020; Sekuboyina et al., 2021), open-access case reports (Europe PMC) (Consortium, 2015), and ∼1,000 de-identified hospital cases. Throughout curation, clinicians (i) defined inclusion criteria and task taxonomies; (ii) vetted imaging selections from hospital cases to prioritize views most informative for diagnosis and surgical planning; and (iii) specified failure modes that instruction data must surface. To minimize hallucinations and preserve traceability, our pipeline (a) extracts figures and text with `PaddleOCR` (Du et al., 2020); (b) *binds images to their local textual context* via caption-pattern regex matching that anchors each figure to its surrounding paragraph; and (c) distills high-quality supervision—multiple-choice, open-ended QA, multi-turn consultations, and report generation—through a *two-stage* LLM process (draft → revision with explicit prompts and logs). Clinicians review and refine prompt policies and revision criteria to align with reporting standards.

**SpineBench** operationalizes evaluation across clinically relevant axes—*imaging report*, *diagnosis*, *patient guidance*, *evidence-based treatment*, *technical feasibility*, *risk prognosis*, *coverage*, *relevance*, *granularity*, and *interpretability*. Its item design, error taxonomy, and rubrics were developed with clinician input to emphasize fine-grained, anatomy-centric reasoning and the kinds of mistakes that matter in practice.

To characterize the state of the field, we evaluate *a dozen* of contemporary large vision–language models (LVLMs) (OpenAI, 2025a;b; Hurst et al., 2024; Google, 2025a;b; Sellergren et al., 2025a; xAI, 2025; Anthropic, 2025; Bai et al., 2025; Hong et al., 2025; Wang et al., 2025a), both general-purpose and medical. Our evaluation reveals significant weaknesses in fine-grained, level-specific diagnosis and open-ended clinical reasoning, particularly in the handling of complex multi-image tasks. Building on these insights, we introduce a fine-tuned spine model SpineGPT trained on SpineMed-450k that delivers consistent improvements on SpineBench as shown in Figure 1. Clinicians assess exemplar outputs for decision relevance, underscoring the practical value of targeted, evidence-linked instruction data. Our contributions are as follows:

- **Clinician-in-the-loop dataset and benchmark.** We release **SpineMed-450k**, more than 450,000 instruction instances spanning multiple-choice, open-ended QA, multi-turn consultations, and report generation—curated via a specialist-supported pipeline with anatomical integration and two-stage report refinement, together with **SpineBench**, a level-aware benchmark co-designed with clinicians and enriched with ∼1,000 real hospital cases.

- **Comprehensive evaluation.** We benchmark *dozens* of open-source LVLMs across closed/open tasks using clinician-shaped taxonomies and rubrics, surfacing systematic failure modes in spine reasoning.

- **A practical baseline model.** We propose a fine-tuned spine LVLM trained on SpineMed-450k that achieves consistent gains on SpineBench; exemplar outputs receive clinician feedback on diagnostic clarity and planning utility, establishing a high-utility baseline for future research.

## 2 SPINEMED-450K DATASET

**Overview.** The SpineMed-450k dataset was constructed through a meticulous "clinician-in-the-loop" pipeline designed to ensure clinical accuracy and relevance. This pipeline integrates four core stages: (1) Dataset collection, (2) Structured Information Extraction, (3) Data De-identification and Cleaning, and (4) Dataset Generation. (5) Annotation of the spinal diagnostic report.

### 2.1 DATA COLLECTION

To build a complete and comprehensive dataset for spinal diagnosis and treatment, we collected data from a variety of sources (Chen et al., 2024a; Wei & Hwei, 2024; Wu et al., 2025; Chen et al., 2024b). Existing general-purpose large vision-language models (Hurst et al., 2024; Google, 2025a;b; Deng et al., 2023; Ullah et al., 2024; AlSaad et al., 2024) and even medical large language models (Li et al., 2023; Wang et al., 2025b; Wu et al., 2024; Lin et al., 2025; Lu et al., 2024; Niu et al., 2025; Nath et al., 2025a; Seyfioglu et al., 2024; Lai et al., 2025; Xu et al., 2025) are trained on generic medical data (Chen et al., 2024a;b; Xie et al., 2024a), which often lacks the high-quality, specialized data needed for orthopedics (Deng et al., 2023; Ullah et al., 2024). To train an effective large model for spinal care, we first compiled a high-quality, general orthopedic dataset covering multiple domains, including Spine Surgery, Foot and Ankle Surgery, Orthopedic Trauma, and Hand and Upper Extremity Surgery.

As shown in Figure 3, we integrated materials from a variety of sources, including textbooks, surgical guidelines, expert consensuses, question banks, open-access case reports from Europe PMC (Consortium, 2015), open single-modality spine datasets (Alibaba Cloud Tianchi, 2020; Sekuboyina et al., 2021) (e.g., Spark, VerSe), and approximately 1,000 de-identified multimodal hospital cases collected from various hospitals. This data covers a wide range of modalities, including text, CT, MRI, X-ray, and tables. We track the provenance (dataset IDs/DOIs, case identifiers) for every derived item. Where possible, we adopt upstream datasets with permissive licenses and clear terms of reuse. Clinicians defined the inclusion criteria and, for hospital cases, selected the most decision-informative images (e.g., MRI target sequences, key CT levels) to serve as the foundation for downstream tasks.

### 2.2 DATASET CURATION

**Structured Information Extraction** To accurately extract comprehensive information from academic sources, we employed PaddleOCR (Du et al., 2020) to parse PDF documents and images from textbooks and literature. The output, containing both recognized text and layout analysis, was

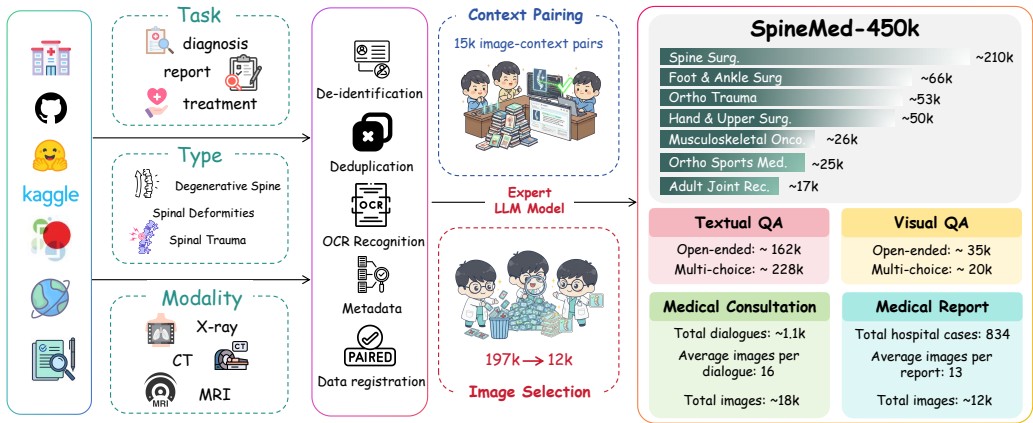

Figure 3: Generation pipeline of SpineMed-450k. The pipeline involves data preprocessing (including de-identification, deduplication, and OCR) followed by expert LLM-driven curation. This process generates 450k items for tasks like QA, medical reports, and consultations across various orthopedic subspecialties.

exported into Markdown format. This approach effectively preserved the structural integrity of the documents, including tables, figure placements, and overall layout. Furthermore, to ensure a precise mapping between figures, their captions, and corresponding contextual descriptions in the text, we developed a novel algorithm termed Picture Context Matching. Subsequently, we employed **GPT-5-mini**(OpenAI, 2025a) to conduct a semantic consistency check, rigorously filtering out instances where the visual content did not align with the retrieved context. The technical details of this algorithm are elaborated in the Appendix E.

**Data De-identification and Cleaning** This stage focused on processing data sourced from a collection of clinical records in hospitals. We first performed a rigorous de-identification process, removing all sensitive and personally identifiable information (PII), such as patient IDs and physical examination details under HIPAA. We also filtered out irrelevant images, such as post-operative photos and non-diagnostic tables. Subsequently, **GPT-5-mini**(OpenAI, 2025a) was utilized to conduct a fine-grained classification of the data, ensuring the dataset's purity by excluding non-orthopedic cases. As shown in Figure 2, the orthopedic domain was categorized into 7 classes, with the spine sub-domain further divided into 14 distinct classes. A detailed statistical overview of the dataset distribution across these categories is presented in Figure 4.

**Dataset Generation** In close collaboration with medical experts, we designed a comprehensive annotation schema to generate high-quality, multi-task training data. The annotation process was tailored to the data source: (1) From External Knowledge Sources (e.g., Textbooks): We generated bilingual (Chinese and English) and multimodal (text and image-based) questions in both multiple-choice and open-ended formats using **Gemini-2.5-pro**(Google, 2025a) with carefully designed prompts. (2) From Opened-spine Datasets: We processed two open-source spinal datasets, Spark and Verse, to generate multi-turn question-and-answer dialogues that simulate doctor-patient interactions. These datasets consist mainly of unimodal 3D image slices (CT and MRI). To ensure consistency, we standardized the inputs by adaptively sampling 25 slices per case under clinical expert supervision. From this, we created over 300 simulated consultations to train models in their conversational abilities within spinal scenarios. (3) From Real Clinical Records: We created multiple-choice questions, multi-turn conversational datasets for patient interviews, and comprehensive spinal diagnostic reports via **locally deployed GLM-4.5V**(Hong et al., 2025) to ensure data security. For prompt design, please refer to the Appendix H

**Annotation of the spinal diagnostic report** A cornerstone of our dataset is the generation of detailed spinal diagnostic reports. In this process, we utilized real clinical reports from hospitals, incorporating physician recommendations, to design reports that encompass six dimensions, all aimed at simulating a complete clinical workflow: (1) Structured Imaging Findings: Analyze the provided medical images and distill key radiological evidence that supports the final diagnosis. (2)

AI-Assisted Diagnosis: Formulate a diagnostic conclusion and articulate the reasoning process based on the synthesis of clinical data and imaging analysis. (3) Treatment Recommendations: This section is bifurcated to address different audiences. Patient-Centric Advice: Explain the rationale for the recommended surgical procedure in clear, non-technical language. Physician-Centric Rationale: Provide a robust, guideline-based decision tree to justify the surgical selection from a clinical perspective. (4) Risk and Prognosis Assessment: Conduct an objective evaluation of the potential risks and expected outcomes associated with the proposed surgical plan. (5) Postoperative Issue Management: Predict potential post-surgical complications for specific procedures and develop corresponding management strategies. (6) Diagnostic Rationale and Disclaimer: Provide complete diagnostic and surgical decision-making chain and disclaimer statement. Report examples are provided in the Appendix G.

## 2.3 Comparison with Existing Benchmarks

Table 1: Comparison of SpineMed-450k with existing spine imaging datasets. While prior datasets focus on specific perception tasks, our work introduces the first multimodal instruction-tuning corpus designed for full-spectrum clinical reasoning.

| Dataset | Modality | Scale | Core Task | Workflow Coverage | Output Format |
|---|---|---|---|---|---|
| RSNA LumbarDISC (Richards et al., 2025) | MRI | 2.6k Patients | Classification | Specific (Stenosis Grading) | Class Labels |
| BUU-LSPINE (Klinwichit et al., 2023) | X-Ray | 3.6k Patients | Detection | Specific (Spondylolisthesis) | Coordinates/Labels |
| VerSe 2020 (Liebl et al., 2021) | CT | 300 Patients | Segmentation | Specific (Anatomy) | Voxel Masks |
| Lumbar Spine MRI (van der Graaf et al., 2024) | MRI | 218 Patients | Segmentation | Specific (Structure) | Voxel Masks |
| Spark (Tianchi) (Tianchi, 2020) | CT, MRI | 150 Patients | Classification | Specific (Disease ID) | Class Labels |
| **SpineMed-450k (Ours)** | **Multimodal** (XR, CT, MRI, Text) | **450k+** Instructions | **Clinical Reasoning** | **Full-Spectrum** (Diag → Treat → Prognosis) | **Instruction Pairs** (Image, Text) |

As illustrated in Table 1, existing datasets such as VerSe(Liebl et al., 2021) and RSNA(Richards et al., 2025) are predominantly unimodal and designed for low-level perception tasks like segmentation, detection, or simple classification. While these datasets serve as effective tools for specific anatomical localization or binary disease identification, they fundamentally fail to capture the holistic context required for complex clinical decision-making, limiting their utility in training models for high-level diagnostic synthesis. In contrast, SpineMed-450k represents a significant paradigm shift from "Tool AI" to "Collaborator AI". Our dataset distinguishes itself through three key dimensions: 1. Multimodal Synthesis, requiring the integration of X-ray, CT, and MRI to mirror real-world cross-modal validation; 2. Cognitive Depth, supporting level-aware reasoning rather than simple label outputs; 3. Workflow Completeness, covering the full patient journey with grounded instructions for Structured Imaging Findings, AI Diagnosis, Treatment Recommendations, and Risk Assessment. This effectively fills the cognitive gap left by perception-focused datasets.

## 3 Data Statistics

SpineMed-450K is a large-scale multimodal training dataset for orthopedic spine knowledge in large language models, characterized by strong traceability, comprehensive coverage, diverse question types, and rich modalities.

### 3.1 Disease Diversity Coverage

As shown in Figure 4(b), SpineMed-450K encompasses seven common orthopedic subspecialties, including Spine Surgery, Foot and Ankle Surgery, and Orthopedic Trauma, with spinal diagnostic data accounting for 47% of the orthopedic data. Furthermore, the spinal diagnostic data includes 14 spine subconditions such as cervical degenerative spine disease and idiopathic scoliosis. We performed sampling on each spinal diagnostic dataset to ensure uniform distribution across all disease categories.

### 3.2 Patient Source Diversity

As illustrated in Figure 4(a), our data originates from 1,000 real clinical cases collected from 11 leading expert hospitals. These data span the recent three years and encompass patients of different genders, various age groups, and diverse physical conditions. To protect privacy, personal information has been de-identified. personal information. Given the varying surgical volumes across different

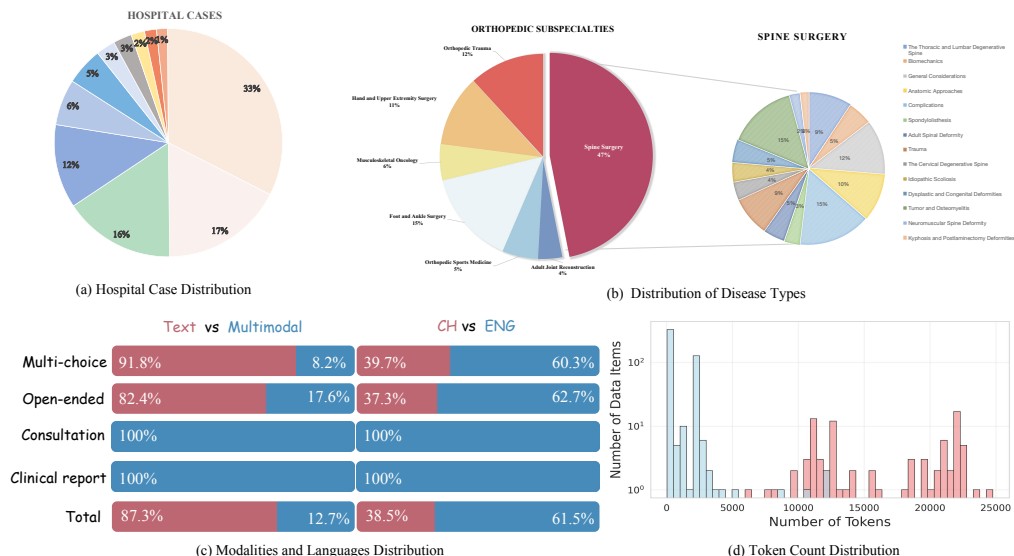

Figure 4: Statistics of SpineMed-450k. (a) Distribution of medical records across various hospitals. (b) The prevalence of various orthopedic and spinal diseases. (c) Distribution of different modals and languages. (d) Benchmark token length distribution: blue (non-report tokens), pink (report tokens).

hospitals, the largest hospital contributes 33% of the data while the smallest contributes 1%. These valuable real patient data provide crucial evidence for accurately representing the authentic conditions of spine patients.

## 3.3 DATA SOURCE AND QUESTION TYPE DIVERSITY

Table 2: Dataset statistics categorized by data source and split.

| Split | Literature | Textbook | Case Report | Question Bank | Open Source | Hospital | Total |
|---|---|---|---|---|---|---|---|
| Train | 6,450 | 377,212 | 61,453 | 1,087 | 304 | 9,668 | 456,174 |
| Test | 17 | 203 | 101 | 3 | – | – | 574 |
| Total | 6,467 | 377,415 | 61,554 | 1,090 | 304 | 9,918 | 456,748 |

As shown in Table 2, our data derives from six major sources: Literature, Textbooks, Case Reports, hospitals, and others. Textbooks, being the primary knowledge source for physicians, constitute the largest proportion

Table 3: Dataset distribution across domains and task types.

| Split | Multiple-choice | Open-ended | Consultation | Report |
|---|---|---|---|---|
| Train | 248,789 | 197,413 | 1,138 | 734 |
| Test | 487 | – | – | 87 |
| Total | 249,276 | 197,413 | 1,138 | 821 |

with 377k entries, while hospital data, though valuable, is limited in quantity, with 9,668 data points generated from nearly 1,000 real cases. As presented in Table 3 and Figure 4(c), question types are categorized into pure text QA, multimodal QA, medical consultations, and clinical reports, with multiple-choice questions comprising the largest proportion. For evaluation convenience, our test set includes only multiple-choice and clinical report formats.

## 3.4 DATA TYPE DIVERSITY

Our dataset incorporates multiple authentic data types including patient physical examination information, patient consultation records, X-rays, CT scans, and MRI images. Due to variations in hospital facilities and patient conditions, the collected data differs for each case, which introduces modeling challenges but enables our trained models to more closely approximate real clinical scenarios faced by physicians.

# 4 SPINEBENCH

## 4.1 BENCHMARK CONSTRUCTION

**Data Sampling** The SpineBench was constructed by sampling from the SpineMed-450k dataset. Following the original distribution of SpineMed-450k, we sampled 500 multiple-choice questions and 100 medical reports. This subset incorporates 14 spinal sub-diseases and data from multiple sources (see Appendix F for details).

**Data Validation** To ensure the integrity of SpineBench, a rigorous review process was implemented involving a team of 17 board-certified orthopedic surgeons. To mitigate bias and ensure objectivity, the surgeons were divided into three independent groups. Each group collaboratively validated the quality of the questions. Erroneous question-answer pairs were corrected, and questions deemed unsuitable for the evaluation set were removed. Ultimately, SpineBench comprises 487 high-quality multiple-choice questions and 87 report generation prompts.

## 4.2 EVALUATION METRICS

Table 4: Evaluation criteria for AI-generated clinical reports across five key dimensions

| Report Section | Evaluation Criterion | Key Assessment Focus |
| --- | --- | --- |
| I. Structured Imaging Report (SIP) | Imaging Report (1-5 pts) | Accuracy of findings, clinical significance, quantitative descriptions |
| II. AI-Assisted Diagnosis (AAD) | Diagnosis (1-5 pts) | Primary diagnosis correctness, differential diagnoses, clinical reasoning |
| III. Treatment Recommendations (TR) | Patient Guidance (1-5 pts)
Evidence-Based Plan (1-5 pts)
Technical Feasibility (1-5 pts) | Language clarity, empathy, patient reassurance
Rationale, individualization, guideline consistency
Surgical details, complication prevention, backup plans |
| IV. Risk & Prognosis Management (RPM) | Risk-Prognosis Mgmt (1-5 pts) | Perioperative planning, follow-up schedule, safety protocols |
| V. Reasoning & Disclaimer (RD) | Coverage (1-5 pts)
Relevance (1-5 pts)
Granularity (1-5 pts)
Explanation (1-5 pts) | Completeness of evidence identification and explanation
Focus on core diagnosis without irrelevant content
Precision and quantitative detail sufficiency
Logical coherence and reasoning chain clarity |

Under the careful design and guidance of our medical team, We propose a comprehensive evaluation framework that integrates three complementary assessment dimensions to measure the overall performance of AI systems in spinal diagnostic tasks:

$$\text{Score}_{\text{total}} = \sum_{k=1}^{3} w_k \cdot P_k \tag{1}$$

where $P_1$, $P_2$, and $P_3$ represent the performance scores for text-only multiple-choice questions, multimodal multiple-choice questions, and diagnostic report generation, respectively. The weights $w_k$ are dynamically determined based on the sample sizes:

$$w_k = \frac{N_k}{\sum_{i=1}^{3} N_i} \tag{2}$$

where $N_k$ denotes the number of samples in each evaluation category. This data-driven weighting scheme ensures statistical reliability while maintaining balanced representation across all assessment dimensions.

The diagnostic report score $P_3$ is computed using our expert-calibrated framework:

$$P_3 = 20 \times \sum_{i=1}^{5} \left( \frac{1}{n_i} \sum_{j=1}^{n_i} s_{ij} \right) \tag{3}$$

where scores are normalized to a 0–100 scale for consistency across all metrics and $s_{ij}$ denotes the score for dimension $j$ in section $i$, $n_i$ represents the number of dimensions in section $i$. This unified scoring system enables direct comparison of model capabilities across diverse clinical tasks, from basic diagnostic reasoning to complex report generation.

## 5 SPINEGPT: A SPECIALIZED CLINICAL COLLABORATOR

In this section, we introduce **SpineGPT** to rigorously validate the efficacy of SpineMed-450k.

### 5.1 IMPLEMENTATION DETAILS

We employed the Qwen2.5-VL-7B-InstructBai et al. (2025) as our foundational architecture. All training phases were executed on a high-performance computational node equipped with 8 NVIDIA A100 GPUs, leveraging the ms-swift framework for efficient fine-tuning. To balance training efficiency with memory constraints across different stages, we dynamically adjusted the DeepSpeed optimization strategies.

Table 5: Training configurations across different stages.

| Configuration | Stage-1 | Stage-2 | Stage-3 |
|---|---|---|---|
| **Datasets** | PubMedVision-150k orthopedics-230k MedThoughts-8K Medical-R1-Distill medical-o1-reasoning | Spine-open Spine-choice | Spine-chat (+reasoning) Spine-report (+reasoning) |
| Learning Rate | 1e-5 | 1e-5 | **1e-6** |
| Max Length | 16,384 | 16,384 | **49,152** |
| DeepSpeed | Zero2 | Zero2 | **Zero3** |
| Epochs | 1 | 1 | 1 |

As detailed in Table 5, for the initial stages involving shorter sequences, we utilized DeepSpeed Zero2; for the final stage requiring extensive context modeling (up to 49k tokens), we transitioned to DeepSpeed Zero3 offloading. The global batch size was optimized per stage to maximize GPU utilization while maintaining training stability.

### 5.2 CURRICULUM LEARNING

This study employs a curriculum learning framework for subsequent training phases, aimed at enhancing the model's applicability and proficiency in the field of orthopedic spine care. The training process is divided into three stages, each integrating distinct datasets and training strategies to progressively strengthen the model's performance in spinal health.

**General and Orthopedic Foundational Learning** In this initial stage, we utilized several publicly available medical text datasets, including medical-o1-reasoning-SFT (Chen et al., 2024a), Medical-R1-Distill-Data (Chen et al., 2024a), and MedThoughts-8K (hw hwei, 2025). Additionally, we incorporated a diverse set of 150,000 multimodal instruction fine-tuning samples uniformly sampled from PubMedVision (Chen et al., 2024b). The primary objective during this phase is to develop the model's foundational capabilities in the medical field and to enhance its performance across various contexts. Subsequently, we trained on data from the SpineMed-450k dataset that pertained to non-spinal categories. Our findings indicate that this non-spinal data significantly improved the model's performance on the SpineBench benchmark, highlighting the importance of broadening the knowledge base to enhance task-specific performance.

**Specialized Learning in Spinal Health** In this phase, we concentrated on all data pertinent to spinal health. Furthermore, we extracted a selection of multiple-choice and open-ended questions to construct long reasoning chains, with the objective of enhancing the model's proficiency in the domain of spinal surgery.

**Enhancement of Report Generation and Conversational Abilities** Finally, we conducted further training through multi-turn dialogues, report generation, and datasets comprising long-chain reasoning instructions. The goal of this stage is to develop the model's advanced language comprehension and generation abilities, particularly in the contexts of dialogue interaction and report creation.

## 6 EXPERIMENTS

### 6.1 COMPARISON AND ANALYSIS ON SPINEBENCH

The evaluation results in Table 6 reveal severe limitations of current vision-language models (OpenAI, 2025a; Hurst et al., 2024; Google, 2025a) in medical domain applications. Large-scale open-source models perform particularly poorly: despite having 72B parameters, Qwen2.5-VL-72B (Bai et al.,

Table 6: Performance comparison of LVLMs on close-ended QA and medical report generation tasks.

| Model | Size | Close-Ended QA | | | Medical Report Generation | | | | | | Avg. |
|---|---|---|---|---|---|---|---|---|---|---|---|
| | | Text | Image | Avg. | SIP | AAD | TR | RPM | RD | Sum | |
| *Proprietary LVLMs* | | | | | | | | | | | |
| GPT5 | - | 87.41 | 79.97 | 84.46 | 4.54 | 4.51 | 4.53 | 4.69 | 4.64 | 91.60 | 85.54 |
| O3 | - | 86.73 | 82.38 | 85.01 | 4.39 | 4.25 | 4.34 | 4.43 | 4.42 | 87.32 | 85.36 |
| Gemini-2.5-Pro | - | 88.44 | **88.60** | **88.50** | **4.55** | **4.51** | **4.68** | 4.79 | **4.80** | 93.32 | **89.23** |
| Claude4 | - | 79.59 | 79.79 | 79.67 | 3.96 | 4.08 | 4.04 | 4.44 | 4.41 | 83.72 | 80.28 |
| GPT-4o | - | 86.73 | 81.70 | 84.74 | 3.16 | 3.03 | 3.06 | 3.30 | 3.46 | 64.04 | 81.60 |
| GPT5-mini | - | 89.12 | 80.83 | 85.83 | 4.55 | 4.48 | 4.60 | **4.98** | 4.78 | **93.56** | 87.01 |
| Gemini-2.5-Flash | - | 83.67 | 80.83 | 82.55 | 4.43 | 4.29 | 4.57 | 4.88 | 4.75 | 91.68 | 83.93 |
| *Open-source LVLMs* | | | | | | | | | | | |
| GLM-4.5V | 21B | 85.71 | 81.35 | 83.98 | 3.85 | 3.78 | 3.83 | 4.05 | 4.30 | 79.24 | 83.26 |
| Qwen2.5-VL-72B | 72B | 84.69 | 79.79 | 82.75 | 3.14 | 3.03 | 3.12 | 3.23 | 3.42 | 63.80 | 79.88 |
| Linshu-32B | 32B | 81.29 | 76.68 | 79.47 | 3.05 | 3.05 | 3.23 | 3.49 | 3.47 | 65.16 | 77.30 |
| Medgemma-27B | 27B | 83.33 | 80.83 | 82.34 | 2.88 | 3.49 | 3.38 | 4.14 | 3.65 | 70.16 | 76.66 |
| HuatuoGPT-7B | 7B | 75.85 | 80.83 | 77.82 | 2.42 | 2.42 | 2.42 | 3.37 | 2.87 | 54.0 | 74.21 |
| Qwen2.5VL-7B | 7B | 75.51 | 74.09 | 74.95 | 2.27 | 2.39 | 2.80 | 3.26 | 2.92 | 54.52 | 64.74 |
| **Ours** | 7B | **89.46** | 84.46 | 87.89 | 4.15 | 4.10 | 4.41 | 4.54 | 4.62 | 87.24 | 87.44 |

2025) achieves only 79.88% average performance and a mere 63.80 cumulative score on medical report generation, far below practical application requirements. Crucially, domain-specific pre-training alone proves insufficient: the medical-specialized Medgemma-27B(Sellergren et al., 2025b) achieves an even lower average of 76.66%, trailing our model by over 10 points despite being nearly 4 times larger. Even the best-performing open-source model GLM-4.5V (Hong et al., 2025) (83.26%) exhibits a nearly 6-point gap compared to the leading proprietary model Gemini-2.5-Pro (89.23%). This gap is more pronounced in medical report generation, where proprietary models exceed 85 points while open-source models struggle to reach 80. Additional medical report results are in the Appendix D.

**Pervasive deficiency in cross-modal alignment.** Nearly all models exhibit varying degrees of performance degradation on multimodal tasks. Among open-source models, GLM-4.5V shows a 4.36-point gap between text (85.71%) and image (81.35%) modalities; Qwen2.5-VL-72B exhibits a 4.90-point gap. Even proprietary models suffer from this issue, with GPT5 dropping from 87.41% on text to 79.97% on images, a gap of 7.44 percentage points. This cross-modal performance disparity reflects fundamental inadequacies in medical image understanding and vision-language alignment in existing models, limiting their application in clinical scenarios requiring comprehensive analysis of medical images and textual information.

**Our method achieves breakthrough performance among open-source models.** We achieve 87.44% average score, outperforming all open-source models by 4.18+ points and exceeding multiple proprietary models on close-ended QA (87.89% vs Claude4's 79.67%, GPT-4o's 84.74%). Our text-only QA (89.46%) surpasses all models including GPT5 (87.41%).

**Efficiency and Clinical Utility.** While we acknowledge that SpineGPT (87.44%) slightly trails the absolute SOTA model, Gemini-2.5-Pro (89.23%), this comparison highlights a remarkable efficiency: our 7B-parameter model uses less than 7% of the parameters compared to Gemini-2.5-Pro (which exceeds 100B parameters), yet achieves ∼98% of its performance. Notably, SpineGPT also outperforms other top-tier models such as GPT-4o (81.60%). This efficiency translates directly to clinical utility, as our lightweight 7B model is safe and efficient enough for local deployment within hospital firewalls, ensuring data privacy without reliance on external cloud APIs.

## 6.2 HUMAN-EXPERT AGREEMENT ANALYSIS

To validate our LLM-based evaluation approach, we conducted a human-expert validation study by sampling cases from our dataset for blind expert scoring. Figure 5 shows the correlation analysis between LLM and expert scores across ten evaluation dimensions. The results demonstrate strong

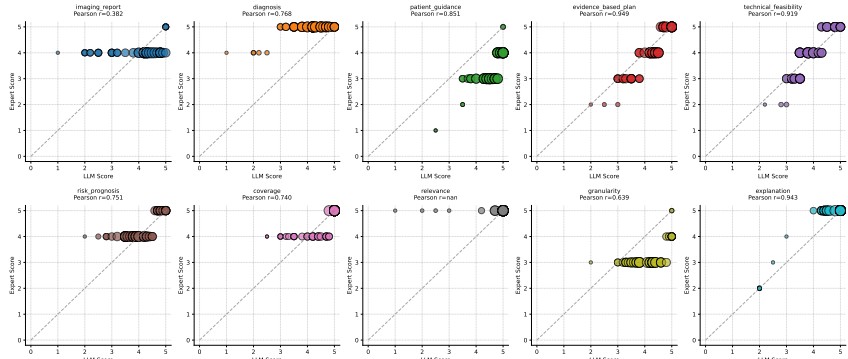

Figure 5: Consistency evaluation of large models and scores given by medical experts

alignment with Pearson correlation coefficients ranging from 0.382 to 0.949, with most dimensions showing correlations above 0.7. These findings validate that our automated LLM scoring serves as a reliable proxy for expert judgment.

## 6.3 ABLATIONS OF SPINEGPT

**Limitations of General Medical Data.** As shown in Table 7, models trained exclusively on large-scale general medical data (row 2) exhibit significant performance degradation (74.95 *vs.* 65.31) on SpineBench compared to the baseline model (row 1). This demonstrates that models trained on such data are insufficient

Table 7: Performance comparison of models on close-ended QA tasks.

| Model | Training Data | | | Close-Ended QA (%) | | |
|---|---|---|---|---|---|---|
| | General | No-Spine | Spine | Text | Image | Avg. |
| Qwen 2.5 VL-7B | | | | 75.51 | 74.09 | 74.95 |
| SpineGPT | ✓ | | | 64.27 | 62.69 | 65.31 |
| SpineGPT | | ✓ | | 82.99 | 80.83 | 82.14 |
| SpineGPT | | | ✓ | 87.76 | 86.01 | 87.07 |
| SpineGPT | ✓ | ✓ | | 83.67 | 77.20 | 81.11 |
| SpineGPT | ✓ | ✓ | ✓ | 89.46 | 84.46 | 87.89 |

for specialized spine diagnostics. The incorporation of the non-spine orthopedic subset derived from our SpineMed-450k corpus (row 3) yields substantial performance improvements (82.14 *vs.* 74.95), validating the importance of domain-aligned training data. Notably, training exclusively on the spine subset (row 4) achieves an impressive average score of 87.07, reaching nearly 99% of the full model's performance. This confirms that our high-density spinal instruction data is the decisive factor. Furthermore, the full multi-stage curriculum (Row 6) incorporates this data to reach the peak performance of 87.89, a substantial enhancement over the configuration relying solely on general medical and orthopedic priors (row 5, 81.11).

# 7 CONCLUSIONS, LIMITATIONS, AND FUTURE WORK

We introduced **SpineMed-450k**, a provenance-rich instruction corpus for level-aware spine diagnosis and planning, and **SpineBench**, level-aware benchmark co-designed with clinicians. Experiments on SpineBench reveal consistent weaknesses of contemporary open-source LVLMs. Our fine-tuned model achieves 87.44% performance, substantially outperforming open-source alternatives and demonstrating that specialized instruction data enables clinically relevant AI capabilities for complex anatomical reasoning tasks.

**Limitations and Future Work.** Future work will expand datasets, train larger models beyond 7B parameters, incorporate reinforcement learning techniques, and provide comprehensive direct comparisons with leading proprietary models including GPT-4 and Gemini to establish clear performance benchmarks.

ACKNOWLEDGEMENTS

This work was supported by the National Key Research and Development Program of China under Grant 2022YFC2407200, and in part by Sanyou Medical, and in part by cash and in-kind funding from Nanjing Kunpeng&Ascend Center of Cultivation and industry partner(s).

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

# APPENDIX

## Contents

# A CHECKLIST

## A.1 ETHICS STATEMENT

This work strictly adheres to the ICLR Code of Ethics. To address potential concerns regarding data provenance, patient privacy, and the redistribution of public or copyrighted materials, we explicitly outline our comprehensive compliance protocols below.

Our data collection strictly adheres to institutional, legal, and copyright frameworks. Patient privacy is guaranteed through rigorous de-identification procedures. Furthermore, to prevent the unauthorized redistribution of copyrighted texts or publicly available benchmark datasets, we employ a "metadata pointer" mechanism and strictly adhere to the principles of transformative use. Specifically, we do not redistribute any raw public medical images, copyrighted paragraphs, original figures, or sensitive clinical records. A detailed breakdown of our data sources, their respective scales, and the specific compliance mechanisms employed is summarized in Table 8.

Table 8: Overview of Data Sources, Scale, and Compliance Mechanisms

| Category | Scale | Source | Compliance & Redistribution Strategy |
|---|---|---|---|
| **Clinical Data** | 9,918 | 11 authorized tertiary orthopedics centers (retrospective) | Zero re-identification risk via 2-stage de-identification (admin info erasure + secure OCR masking). **No Raw Redistribution:** Only a representative de-identified subset is released under Controlled Access requiring a Data Use Agreement (DUA). |
| **Open Datasets** | 304 | Public spine imaging benchmarks (VerSe, Spark) | **No Image Redistribution:** Strict adherence to original terms. We release only instruction annotations via a *Metadata Pointer* mechanism, requiring users to map our scripts to original images downloaded directly from official repositories. |
| **Literature & Reports** | 68,021 | Europe PMC Open Access subset | **API Access:** Gathered via official RESTful API. Strict filtering applied for Open Access licenses that explicitly permit academic text and data mining. |
| **Textbooks & Guidelines** | 377,415 | Renowned orthopedics textbooks & top Spine journals | **Transformative Use:** Extracted solely for factual medical concepts, diagnostic logic, and reasoning chains to guide LLM synthesis. Absolutely no copying or redistribution of original copyrighted text or figures. |
| **Question Banks** | 1,090 | Publicly accessible medical exams | **Transformative Use:** Utilized purely as structural templates for MCQ reasoning. The final output consists entirely of LLM-generated synthetic data, constituting academic fair use. |

## A.2 REPRODUCIBILITY STATEMENT

To maximize transparency and guarantee reproducibility while strictly adhering to institutional data governance and copyright laws, we adopt a tiered open-source release strategy. We will fully open-source the SpineBench evaluation framework, all codebase implementations (training, inference, and evaluation scoring), and the complete SpineMed-450k instruction annotations (JSON files comprising QA pairs and reasoning chains). To prevent the unauthorized redistribution of copyrighted source images from public benchmarks (e.g., VerSe, Spark), we do not release the raw image files. Instead, we employ a "Metadata-Pointer" mechanism, providing explicit data indices that allow researchers to map our open-source instructions to the original images acquired directly from their official repositories. Regarding the highly sensitive proprietary hospital data, full open-sourcing is legally prohibited. However, to ensure the verifiability of our claims, we will release a representative, fully de-identified clinical subset under a Controlled Access model governed by a Data Use Agreement

(DUA). This released subset, in conjunction with the public data pointers, provides the exact necessary resources to fully reproduce all SpineBench evaluation metrics reported in this paper.

### A.3 LLM USAGE

Large Language Models (LLMs) were used to aid in the writing and polishing of the manuscript. Specifically, we used an LLM to assist in refining the language, improving readability, and ensuring clarity in various sections of the paper. The model helped with tasks such as sentence rephrasing, grammar checking, and enhancing the overall flow of the text.

It is important to note that the LLM was not involved in the ideation, research methodology, or experimental design. All research concepts, ideas, and analyses were developed and conducted by the authors. The contributions of the LLM were solely focused on improving the linguistic quality of the paper, with no involvement in the scientific content or data analysis.

The authors take full responsibility for the content of the manuscript, including any text generated or polished by the LLM. We have ensured that the LLM-generated text adheres to ethical guidelines and does not contribute to plagiarism or scientific misconduct.

## B CONTRIBUTIONS AND ACKNOWLEDGMENTS

1. **Ming Zhao**, $\pi^3$ Lab, Jilin University
2. **Wenhui Dong**, $\pi^3$ Lab, Nanjing University
3. **Yang Zhang**, The Fourth Medical Center of People's Liberation Army General Hospital
4. **Xiang Zheng**, $\pi^3$ Lab, Institute of Automation, Chinese Academy of Sciences
5. **Zhonghao Zhang**, $\pi^3$ Lab, Ningxia University
6. **Zi An Zhou**, $\pi^3$ Lab, Zhejiang University
7. **Yunzhi Guan**, Huashan Hospital, Fudan University
8. **Liukun Xu**, The Fourth Medical Center of People's Liberation Army General Hospital
9. **Wei Peng**, $\pi^3$ Lab
10. **Zhaoyang Gong**, Huashan Hospital, Fudan University
11. **Zhicheng Zhang**, The Fourth Medical Center of People's Liberation Army General Hospital
12. **Dachuan Li**, Huashan Hospital, Fudan University
13. **Xiaosheng Ma**, Huashan Hospital, Fudan University
14. **Yuli Ma**, Sanyou Medical
15. **Jianing Ni**, $\pi^3$ Lab
16. **Changjiang Jiang**, Wuhan University
17. **Lixia Tian**, Beijing Jiaotong University
18. **Qixin Chen**, The Second Affiliated Hospital, Zhejiang University
19. **Kaishun Xia**, The Second Affiliated Hospital, Zhejiang University
20. **Pingping Liu**, Jilin University
21. **Tongshun Zhang**, Jilin University
22. **Zhiqiang Liu**, $\pi^3$ Lab, Huazhong University of Science and Technology
23. **Zhongan Bi**, $\pi^3$ Lab, Zhejiang University
24. **Chenyang Si**, Nanjing University
25. **Tiansheng Sun**, The Fourth Medical Center of People's Liberation Army General Hospital
26. **Caifeng Shan**, Nanjing University

## C RELATED WORK

The landscape of medical AI is rapidly evolving, moving from broad, general-purpose models to highly specialized systems designed for clinical utility. Our work is situated within this trend, addressing a critical gap in the high-stakes field of spine surgery.

**From Generalist Models to Domain Adaptation.** Recent advances in Large Vision-Language Models (LVLMs), such as GPT-4V (OpenAI, 2023) and Gemini2.5-pro (Google, 2025a), have demonstrated significant progress in multimodal tasks including image and video(Yang et al., 2023; Team et al., 2023; Zheng et al., 2023; Jiang et al., 2025; Guo et al., 2025a; Zhu et al., 2024; Jiang et al., 2026). However, when applied to the medical domain, their generalist nature becomes a

distinct limitation. Multiple evaluations consistently show that while promising, these models lack the domain-specific expertise required for complex diagnostic tasks, performing below the level of human specialists (AlSaad et al., 2024). This inherent limitation of generalist models has fueled a clear and necessary trend toward specialization. In response, specialized medical LVLMs like LLaVA-Med (Li et al., 2023) and PMC-LLaMA (Wu et al., 2024) have been developed, fine-tuned on large biomedical corpora. Nevertheless, this approach still has shortcomings. For instance, in spinal diagnostics, a critical task is the synthesis of data from multimodal imaging—such as X-ray, CT, and MRI—to formulate a single, "level-aware" diagnosis. This integrative reasoning process, which requires localizing findings to specific vertebral levels, is a clinical skill that cannot be acquired from static, descriptive datasets alone. This further underscores a core principle: for high-stakes clinical applications, deep, narrow expertise is far more valuable than broad, superficial general knowledge. A powerful example validating this principle is OralGPT (Hao et al., 2025), a model trained on a small, highly curated dataset of intraoral photographs, which achieves performance comparable to state-of-the-art generalist models within its niche. This paradigm shift from generalist to specialist models is now clearly evident across numerous medical fields, from oncology to pathology (Qiu et al., 2023; Sarabadani et al., 2025; Yang et al., 2025; 2024; Barrit et al., 2024; Mo et al., 2025; Deng et al., 2024; Xue et al., 2024; Bhaumik et al., 2023; Na, 2024; Guo et al., 2025b).

**Foundational Datasets and the Cognitive Gap.** Progress in AI is fundamentally tied to the quality of training data. Foundational datasets like MIMIC-CXR (Johnson et al., 2019) and CheXpert (Irvin et al., 2019) have been instrumental for tasks like chest radiograph classification. Moving up in complexity are datasets for interactive Visual Question Answering (VQA). For instance, VQA-RAD (Lau et al., 2018) was manually constructed by clinicians asking naturally occurring questions about radiology images, representing a step toward more dynamic reasoning. More recently, large-scale efforts like MedTrinity-25M (Xie et al., 2024b) have emerged, providing over 25 million images with multi-granular annotations to support a wide range of tasks.

Within the spine domain itself, public datasets have primarily supported foundational computer vision tasks. As summarized in Table 1, existing datasets largely focus on single-modality perception tasks. For example, the RSNA LumbarDISC dataset (Richards et al., 2025) provides a large-scale MRI benchmark but is limited to the classification of stenosis severity grades. Similarly, BUU-LSPINE (Klinwichit et al., 2023) focuses on spondylolisthesis detection in X-rays, while the VerSe 2020 (Liebl et al., 2021) and Lumbar Spine MRI (van der Graaf et al., 2024) datasets provide voxel-level masks for segmentation tasks in CT and MRI, respectively.

However, these resources are primarily designed to support lower-level cognitive tasks like perception ("Where is the L4 vertebra?") or classification ("Is a fracture present?"). They lack the multimodal integration and instruction-following structure required to train models for the highest level of clinical cognition: synthesizing multimodal information into a comprehensive diagnosis and treatment plan. This reveals a crucial gap between existing data paradigms and the needs of clinical practice—a gap our work aims to fill by introducing the first large-scale instruction-tuning corpus designed for full-spectrum clinical reasoning.

**AI in Spine Analysis: From Tools to Collaborators.** Prior AI applications in spine analysis have focused on discrete tasks, creating valuable "tools" rather than "collaborators." These include automated vertebral segmentation and the measurement of spinal parameters (Lee et al., 2024a; Ibrahim et al., 2025a). While useful for improving efficiency, these tools perform isolated tasks, leaving the cognitive burden of synthesis and planning to the human clinician (Nath et al., 2025b). Our work directly addresses these gaps. By creating SpineMed-450k, a large-scale dataset derived from clinical workflows, and SpineBench, a benchmark focused on level-aware, multimodal reasoning, we provide the infrastructure to build and evaluate AI systems that can function as true clinical collaborators in the complex domain of spine surgery.

# D   PERFORMANCE COMPARISON ON MEDICAL REPORT GENERATION SUBTASKS

Table 9: LVLM performance comparison on medical report generation subtasks: Imaging Report (IR), Diagnosis (DGN), Patient Guidance (PG), Evidence-Based Plan (EBP), Technical Feasibility (TF), Risk Prognosis Management (RPM), Coverage (COV), Relevance (REL), Granularity (GRA), Explanation (EXP).

| Model | IR | DGN | PG | EBP | TF | RPM | COV | REL | GRA | EXP |
|---|---|---|---|---|---|---|---|---|---|---|
| *Proprietary LVLMS* | | | | | | | | | | |
| GPT-5 | 4.54 | 4.51 | 4.62 | 4.41 | 4.56 | 4.69 | 4.58 | 4.66 | 4.74 | 4.60 |
| O3 | 4.39 | 4.25 | 4.32 | 4.30 | 4.40 | 4.43 | 4.34 | 4.45 | 4.50 | 4.39 |
| Gemini-2.5-Pro | **4.55** | **4.51** | **4.79** | **4.60** | 4.64 | 4.79 | **4.69** | 4.83 | 4.84 | **4.80** |
| Claude-4 | 3.96 | 4.08 | 4.41 | 3.76 | 3.94 | 4.44 | 4.30 | 4.58 | 4.62 | 4.16 |
| GPT-4o | 3.16 | 3.03 | 3.30 | 3.07 | 2.80 | 3.30 | 3.35 | 4.30 | 2.92 | 3.25 |
| GPT-5-mini | 4.55 | 4.48 | 4.62 | 4.47 | **4.71** | **4.98** | 4.66 | 4.87 | **4.90** | 4.67 |
| Gemini-2.5-Flash | 4.43 | 4.29 | 4.73 | 4.51 | 4.48 | 4.88 | 4.64 | **4.89** | 4.82 | 4.67 |
| *Open-source LVLMS (>10B)* | | | | | | | | | | |
| GLM-4.5V | 3.85 | 3.78 | 4.12 | 3.77 | 3.59 | 4.05 | 4.26 | 4.63 | 4.23 | 4.09 |
| Qwen2.5-VL-72B | 3.14 | 3.03 | 3.25 | 3.09 | 3.02 | 3.23 | 3.27 | 4.19 | 2.98 | 3.25 |
| LinShu-32B | 3.05 | 3.05 | 3.22 | 3.44 | 3.04 | 3.49 | 3.21 | 4.34 | 2.90 | 3.44 |
| Medgemma-27B | 2.88 | 3.49 | 4.14 | 3.56 | 3.32 | 3.26 | 3.48 | 4.29 | 3.51 | 3.32 |
| *Open-source LVLMS (<10B)* | | | | | | | | | | |
| HuaTuoGPT-7B | 2.42 | 2.42 | 2.91 | 2.76 | 2.77 | 3.37 | 2.77 | 3.50 | 2.57 | 2.63 |
| Qwen2.5-VL-7B | 2.27 | 2.39 | 2.82 | 2.86 | 2.71 | 3.26 | 2.77 | 3.66 | 2.60 | 2.65 |
| **Ours** | 4.15 | 4.10 | 4.71 | 4.27 | 4.25 | 4.54 | 4.51 | 4.81 | 4.58 | 4.53 |

# E   PICTURE CONTEXT MATCHING ALGORITHM

The following algorithm processes Markdown files to extract image information and generate structured metadata in JSON format through parallel processing.

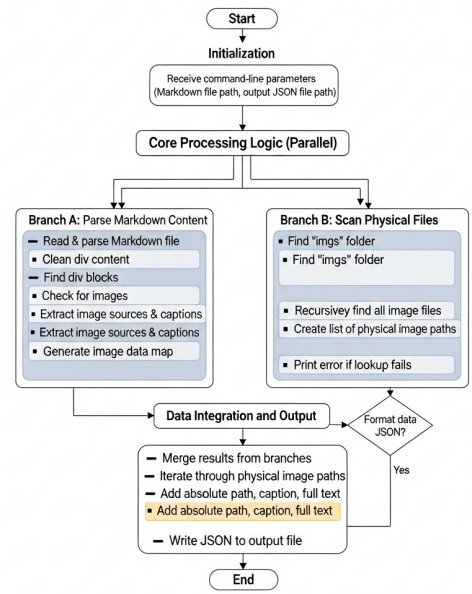

Figure 6: picture context matching algorithm

# F DETAILED DISTRIBUTION ANALYSIS OF SPINEBENCH

To further validate the clinical representativeness and unbiased nature of SpineBench, we provide a comprehensive statistical breakdown of the testing QA set (487 items with specific subspecialty tags). These statistics demonstrate that SpineBench achieves rigorous coverage across diverse pathologies and data sources.

## F.1 DISEASE SUBSPECIALTY DISTRIBUTION

As shown in Table 10, the benchmark covers 14 distinct spinal subspecialties. Notably, high-stakes and complex conditions such as *Tumor and Osteomyelitis* (16.72%) and *Complications* (10.80%) are heavily represented, ensuring that the evaluation reflects performance on critical clinical challenges rather than being dominated by common degenerative cases.

Table 10: Detailed Distribution of Spine Subspecialties in SpineBench

| Rank | Subspecialty | Count | Percentage |
|---|---|---|---|
| 1 | Tumor and Osteomyelitis | 96 | 16.72% |
| 2 | Complications | 62 | 10.80% |
| 3 | Dysplastic and Congenital Deformities | 49 | 8.54% |
| 4 | Idiopathic Scoliosis | 47 | 8.19% |
| 5 | The Thoracic and Lumbar Degenerative Spine | 34 | 5.92% |
| 6 | General Considerations | 33 | 5.75% |
| 7 | Kyphosis and Postlaminectomy Deformities | 31 | 5.40% |
| 8 | Anatomic Approaches | 31 | 5.40% |
| 9 | Adult Spinal Deformity | 26 | 4.53% |
| 10 | Spondylolisthesis | 23 | 4.01% |
| 11 | Trauma | 23 | 4.01% |
| 12 | Neuromuscular Spine Deformity | 12 | 2.09% |
| 13 | Biomechanics | 11 | 1.92% |
| 14 | The Cervical Degenerative Spine | 9 | 1.57% |
| **Total** | **All Subspecialties** | **487** | **100.00%** |

## F.2 DATA SOURCE DIVERSITY

Table 11 illustrates the provenance of the testing data. The dataset maintains a rigorous balance between *Academic Resources* (45.8%), which ensure theoretical precision, and *Real-world Clinical Data* (54.2%), which test practical diagnostic robustness. Crucially, the Hospital Databases (33.5%) consist of private, internal cases that ensure a zero-leakage evaluation environment.

Table 11: Detailed Source Distribution of SpineBench QAs

| Category | Specific Composition | Count | % | Evaluation Goal |
|---|---|---|---|---|
| **Textbooks** | Standard Medical Textbooks | 203 | 41.7% | Theoretical Foundation |
| **Hospital Databases** | Proprietary data related to privacy | 163 | 33.5% | Surgical Precision & Zero-Leakage |
| **Clinical Case Reports** | Real-world Case Reports | 101 | 20.7% | Complex Clinical Scenarios |
| **Literature** | Academic Papers | 17 | 3.5% | Cutting-edge Knowledge |
| **Question Bank** | Medical Exam Questions | 3 | 0.6% | Standardized Logic |
| **Total** | | **487** | **100%** | |

# G   QUANTITATIVE COMPARISON OF SPINEGPT WITH GPT-4O

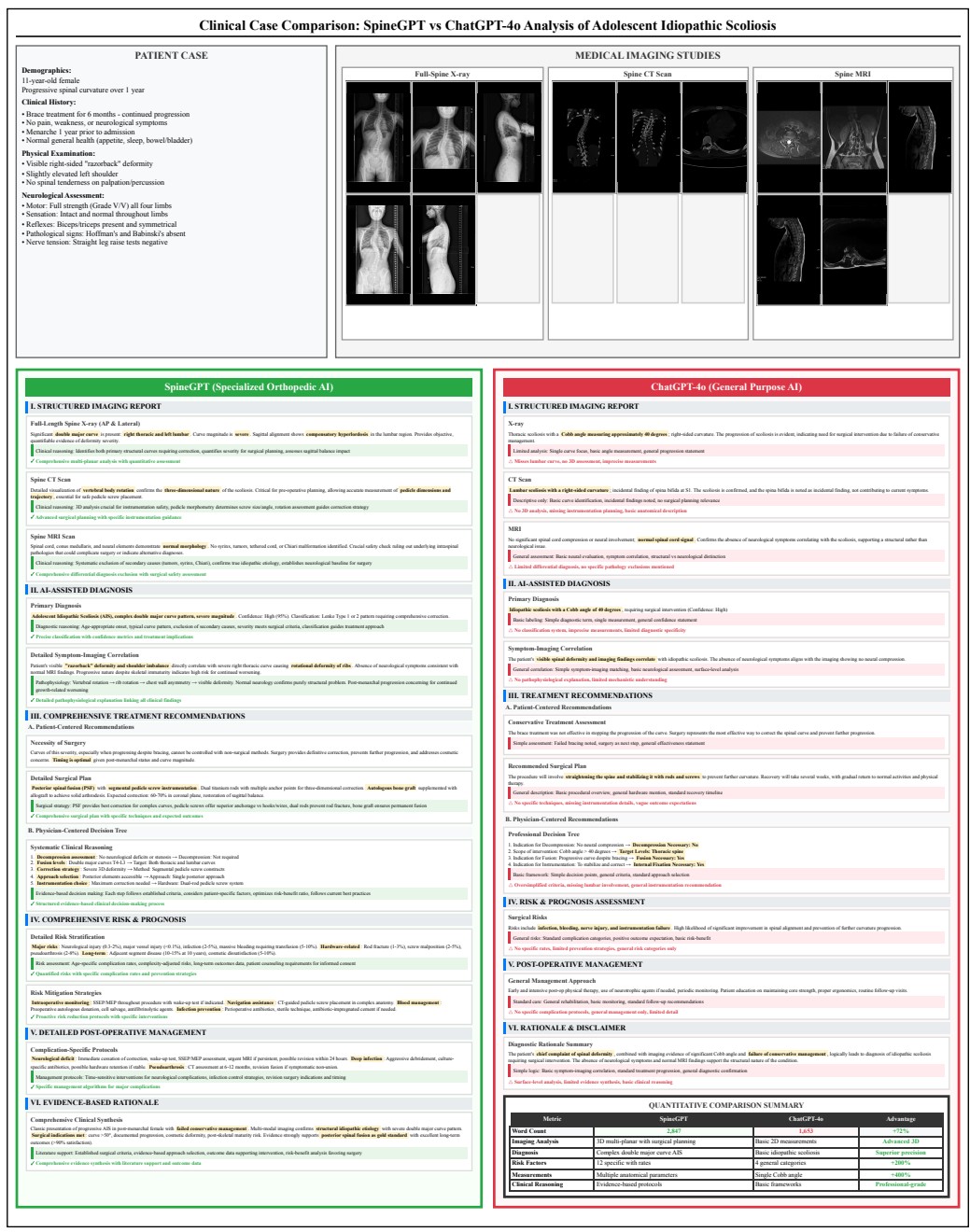

Figure 7: Comparative analysis of medical report generation capabilities between SpineGPT (Ours) and ChatGPT-4o (general-purpose AI) for an adolescent idiopathic scoliosis case. The comparison demonstrates significant differences in diagnostic depth, clinical reasoning, and treatment planning specificity. SpineGPT provides 72protocols, while ChatGPT-4o offers basic diagnostic and treatment recommendations suitable for general medical documentation.

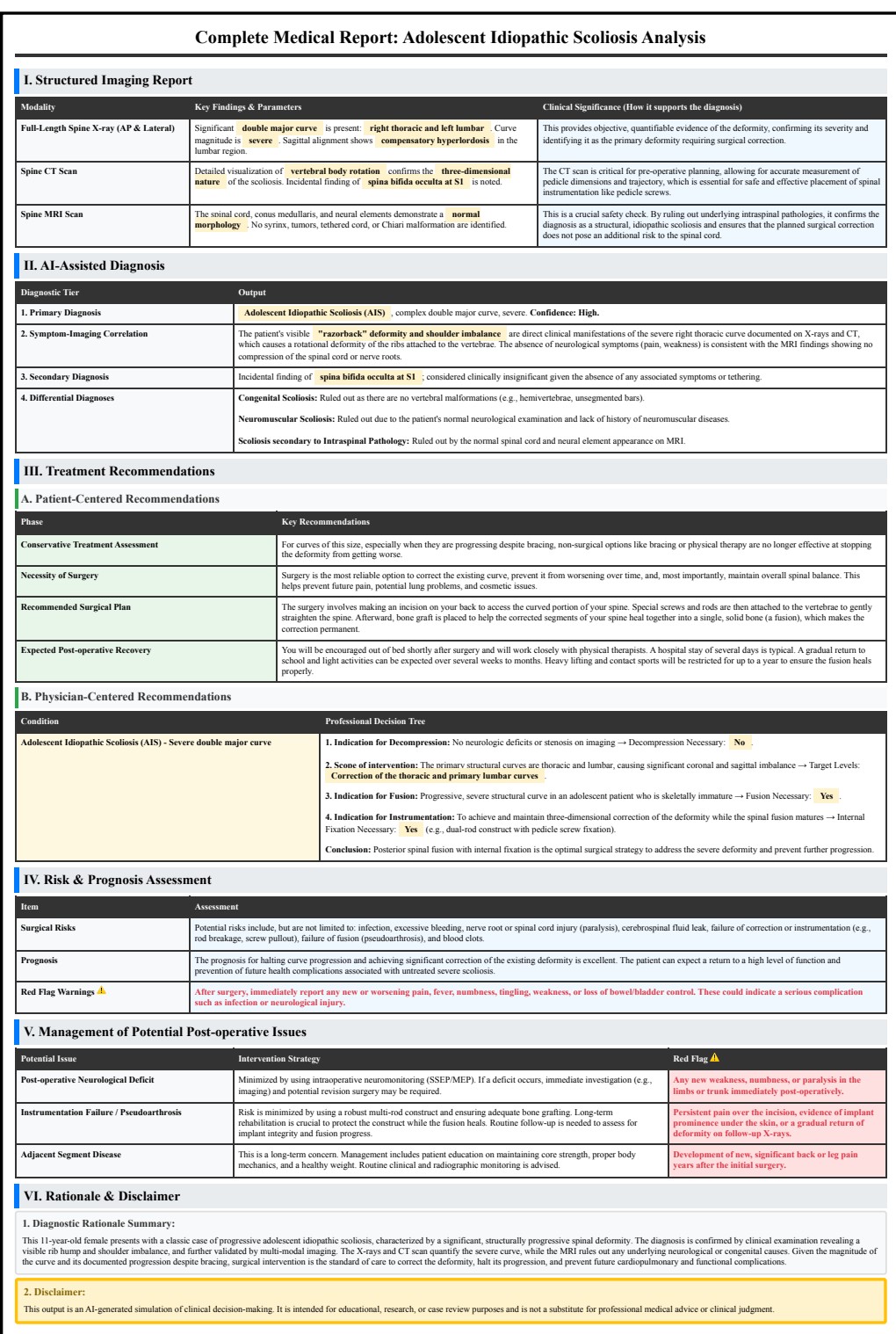

Figure 8: Our model's medical report output for adolescent idiopathic scoliosis, featuring six-section structured format: imaging findings, AI diagnosis, treatment recommendations, risk assessment, post-operative management, and clinical rationale.

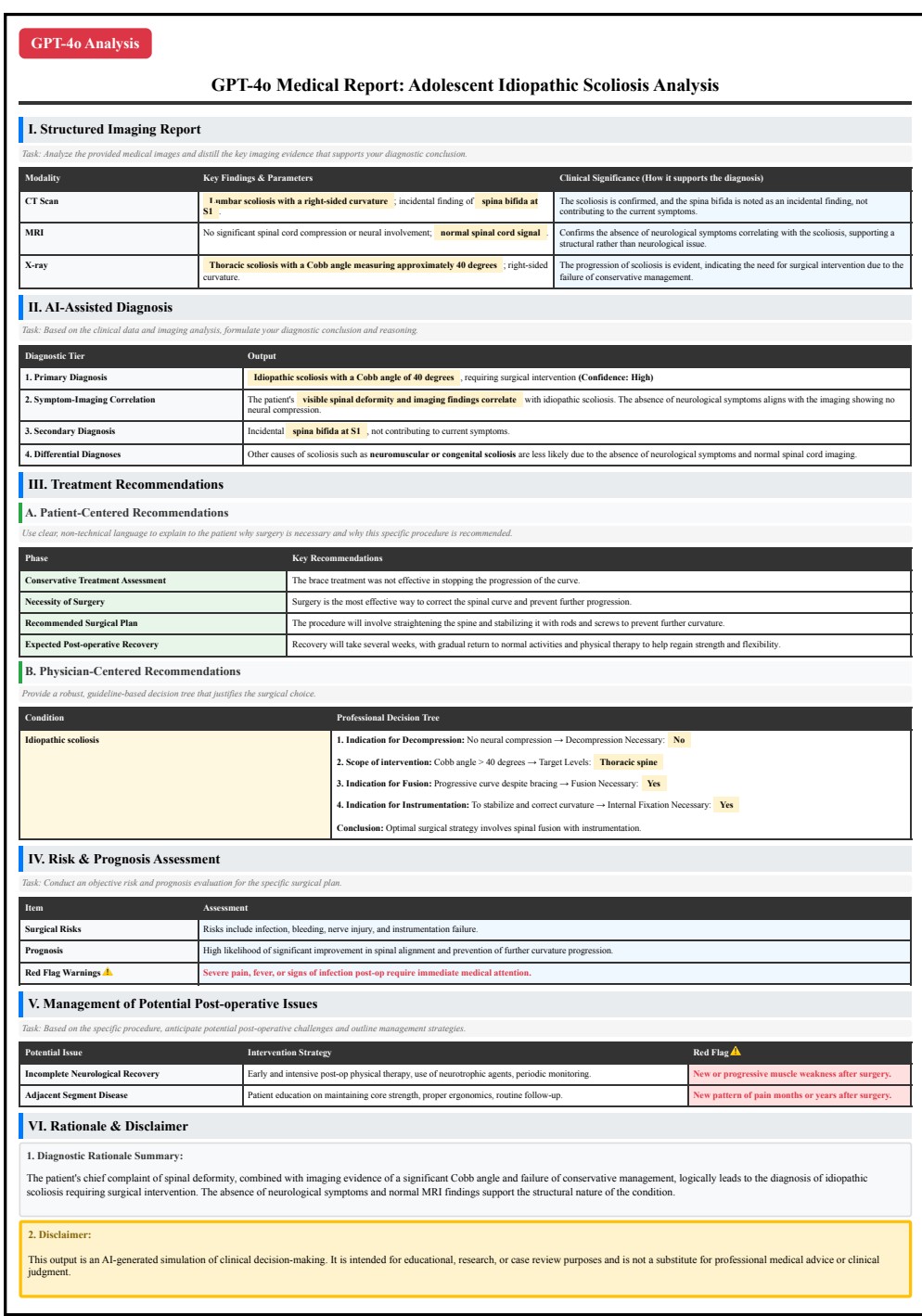

Figure 9: ChatGPT-4o generated medical report for adolescent idiopathic scoliosis, showing general-purpose AI's approach to clinical documentation with basic diagnostic and treatment recommendations.

# H PROMPTS

---

**Criteria for Assessing Dimensional Quality in Reports**

**I. Role and Core Task**
You will act as a **top-tier clinical medical expert** and **AI evaluator** (LLM-Judge). Your core task is to rigorously compare the **[LLM Generated Answer]** provided below with the **[Standard Answer]**. Based on a comprehensive and detailed scoring rubric, you will systematically evaluate the **[LLM Generated Answer]**'s performance against the **[Standard Answer]** across multiple dimensions, including **accuracy, completeness, logical coherence, readability,** and **clinical utility**. Finally, you will output your evaluation in the specified simple format.

**II. Inputs for Evaluation**
1. **[Standard Answer] (Golden Answer)**
**[Please paste the ideal, golden standard answer here]**
2. **[LLM Generated Answer]**
**[Please paste the AI-generated answer that requires evaluation here]**

**III. Evaluation Instructions & Scoring Rubric**
Please score the **[LLM Generated Answer]** for each dimension below, strictly based on its comparison with the **[Standard Answer]**.
**Please note:** Scores MUST be continuous values (e.g., 3.5, 4.2, 4.7) to more precisely reflect the subtle differences in the evaluation results between the integer standards. The integer scores (1-5 points) in the rubric should serve as the primary anchors for your scoring, but you should use decimal precision to capture nuanced differences. For example:
- If performance is slightly above **"Good"** (4 pts) but not quite **"Excellent"** (5 pts), use scores like 4.3 or 4.6.
- If performance has minor gaps compared to **standard**, use scores like 3.8 or 4.2.
- Avoid whole numbers unless the performance exactly matches the integer anchor description.

1. **Structured Imaging Report**
**5 pts (Excellent / On Par)**: On par with the **[Standard Answer]**, accurately describes all key imaging findings, correctly explains their clinical significance, and includes quantitative descriptions.
**4 pts (Good / Minor Gaps)**: The description of major findings is correct, but it lacks some of the quantitative details present in the **[Standard Answer]**.
**3 pts (Fair / Clear Gaps)**: The description is generally correct, but the explanation of clinical significance is clearly less sufficient or in-depth than the **[Standard Answer]**.
**2 pts (Poor / Serious Deficiencies)**: Omits or incorrectly describes key findings that are mentioned in the **[Standard Answer]**.
**1 pt (Unacceptable / Completely Wrong)**: Seriously misinterprets the imaging, with key conclusions that contradict the **[Standard Answer]**.

2. **AI-Assisted Diagnosis**
**5 pts (Excellent / On Par)**: On par with the **[Standard Answer]**, the primary diagnosis is completely correct, secondary diagnoses are reasonably listed, and key differential diagnoses are correctly ruled out.
**4 pts (Good / Minor Gaps)**: The primary diagnosis is correct, but the list of secondary diagnoses is less complete than in the **[Standard Answer]**.
**3 pts (Fair / Clear Gaps)**: The primary diagnosis is correct but omits important differential diagnoses that are mentioned in the **[Standard Answer]**.
**2 pts (Poor / Serious Deficiencies)**: The primary diagnosis is partially incorrect or omits key components present in the **[Standard Answer]**.
**1 pt (Unacceptable / Completely Wrong)**: The diagnosis is completely wrong or misses a life-threatening condition.
3. **Treatment Recommendations**
3.1 **Patient-Oriented Advice**
**5 pts (Excellent / On Par)**: On par with the **[Standard Answer]**, the language is extremely colloquial and easy to understand, the information is completely accurate, the structure is clear, and it is highly effective at reassuring the patient.

Figure 10: Criteria for Assessing Dimensional Quality in Reports

---

### Criteria for Assessing Dimensional Quality in Reports

**4 pts (Good / Minor Gaps)**: The language is easy to understand and the core information is accurate, but the level of empathy or nuance is slightly inferior to the **[Standard Answer]**.

**3 pts (Fair / Clear Gaps)**: The language is generally understandable but contains unexplained jargon, the information is mostly correct but vague, and it clearly lacks the empathy shown in the **[Standard Answer]**.

**2 pts (Poor / Serious Deficiencies)**: The language is obscure and jargon-heavy, the information contains errors or critical omissions, and it is likely to cause patient anxiety.

**1 pt (Unacceptable / Completely Wrong)**: The communication contains serious errors, is misleading, or provides harmful information, making it completely unacceptable.

3.2 **Treatment Plan & Evidence-Based Consistency**

**5 pts (Excellent / On Par)**: The plan's rationale, individualization, and discussion of evidence-based support are all on par with the depth and breadth of the **[Standard Answer]**.

**4 pts (Good / Minor Gaps)**: The core plan is reasonable, but the discussion of the evidence base is less detailed than in the **[Standard Answer]**.

**3 pts (Fair / Clear Gaps)**: The plan is generally reasonable but lacks the individualized adjustments highlighted in the **[Standard Answer]**.

**2 pts (Poor / Serious Deficiencies)**: Parts of the plan are inconsistent with clinical guidelines, making its rationale far weaker than the **[Standard Answer]**'s.

**1 pt (Unacceptable / Completely Wrong)**: The plan clearly conflicts with evidence-based medicine and is diametrically opposed to the recommendations in the **[Standard Answer]**.

3.3 **Surgical/Technical Details & Feasibility**

**5 pts (Excellent / On Par)**: The explanation of surgical goals, technical details, preventive measures, and backup plans is comparable in completeness and professionalism to the **[Standard Answer]**.

**4 pts (Good / Minor Gaps)**: Covers the main technical details, but its consideration of complication prevention is less thorough than the **[Standard Answer]**'s.

**3 pts (Fair / Clear Gaps)**: The description of details is overly general and lacks the specificity and feasibility assessment present in the **[Standard Answer]**.

**2 pts (Poor / Serious Deficiencies)**: Omits key technical details mentioned in the **[Standard Answer]**, making its feasibility questionable.

**1 pt (Unacceptable / Completely Wrong)**: The technical details are infeasible or pose a safety risk.

4. **Risk, Prognosis & Post-Op Management**

**5 pts (Excellent / On Par)**: Provides a perioperative management plan, follow-up schedule, and strategy for potential issues that is as systematic, complete, and forward-thinking as the **[Standard Answer]**.

**4 pts (Good / Minor Gaps)**: Covers the main measures but is less systematic or detailed in certain aspects compared to the **[Standard Answer]**.

**3 pts (Fair / Clear Gaps)**: Mentions basic safety measures but lacks the systematic and structured approach demonstrated in the **[Standard Answer]**.

**2 pts (Poor / Serious Deficiencies)**: Omits important safety protocols that are emphasized in the **[Standard Answer]**.

**1 pt (Unacceptable / Completely Wrong)**: Seriously neglects safety, contradicting the patient-centric principles of the **[Standard Answer]**.

5. **Theoretical Basis & Disclaimer (EVA 4D Evaluation)**

5.1 **Coverage**

**5 pts (Excellent / On Par)**: On par with the **[Standard Answer]**, accurately identifies and explains all key pieces of evidence with no omissions.

**4 pts (Good / Minor Gaps)**: Covers most key evidence but may omit one non-critical element that was included in the **[Standard Answer]**.

**3 pts (Fair / Clear Gaps)**: Covers the main evidence but omits one key element or two minor elements present in the **[Standard Answer]**.

**2 pts (Poor / Serious Deficiencies)**: Covers only a small amount of evidence, and the chain of reasoning is far less complete than the **[Standard Answer]**'s.

**1 pt (Unacceptable / Completely Wrong)**: Fails to cover any key evidence, or the evidence cited contradicts the factual basis of the **[Standard Answer]**.

Figure 11: Criteria for Assessing Dimensional Quality in Reports

---

**Criteria for Assessing Dimensional Quality in Reports**

5.2 **Relevance**
**5 pts (Excellent / On Par)**: On par with the **[Standard Answer]**, all discussion is tightly focused on the core diagnosis and decision, with no irrelevant content.
**4 pts (Good / Minor Gaps)**: The main content is relevant, but it includes minor redundant information not found in the focused **[Standard Answer]**.
**3 pts (Fair / Clear Gaps)**: The discussion mixes relevant and irrelevant information, diluting the focus compared to the **[Standard Answer]**.
**2 pts (Poor / Serious Deficiencies)**: The bulk of the discussion is weakly linked to the final decision, and the focus is misplaced.
**1 pt (Unacceptable / Completely Wrong)**: The discussion is entirely irrelevant to the diagnosis or is based on incorrect assumptions.

5.3 **Granularity**
**5 pts (Excellent / On Par)**: On par with the **[Standard Answer]**, provides precise, quantitative details sufficient to support in-depth clinical judgment.
**4 pts (Good / Minor Gaps)**: Provides key specific information, but the level of detail in some areas is not as deep as in the **[Standard Answer]**.
**3 pts (Fair / Clear Gaps)**: The information is overly general and lacks the distinguishing details found in the **[Standard Answer]**.
**2 pts (Poor / Serious Deficiencies)**: Uses highly generalized language, providing far less informational value than the **[Standard Answer]**.
**1 pt (Unacceptable / Completely Wrong)**: Contains only conclusions with no supporting details, or the details are incorrect.

5.4 **Explanation**
**5 pts (Excellent / On Par)**: On par with the **[Standard Answer]**, the chain of reasoning is clear, complete, and seamless, with all parts logically supporting the conclusion.
**4 pts (Good / Minor Gaps)**: The overall logic is coherent, but the reasoning for a specific step is slightly less clear or direct than in the **[Standard Answer]**.
**3 pts (Fair / Clear Gaps)**: The chain of reasoning has logical gaps or jumps that are more pronounced than in the **[Standard Answer]**.
**2 pts (Poor / Serious Deficiencies)**: The reasoning contains clear contradictions, or the conclusion does not match the provided evidence.
**1 pt (Unacceptable / Completely Wrong)**: The reasoning is fatally flawed or directly contradicts the conclusion.

**IV. Required Output Format**
Please strictly follow this simple format. Each line should contain exactly one score and justification:
**IMAGING_REPORT: [Score] | [Justification]**
**DIAGNOSIS: [Score] | [Justification]**
**PATIENT_GUIDANCE: [Score] | [Justification]**
**EVIDENCE_BASED_PLAN: [Score] | [Justification]**
**TECHNICAL_FEASIBILITY: [Score] | [Justification]**
**RISK_PROGNOSIS: [Score] | [Justification]**
**COVERAGE: [Score] | [Justification]**
**RELEVANCE: [Score] | [Justification]**
**GRANULARITY: [Score] | [Justification]**
**EXPLANATION: [Score] | [Justification]**

**Example:**
**IMAGING_REPORT: 4.2 | The report accurately describes key findings but lacks some quantitative details.**
**DIAGNOSIS: 3.8 | Primary diagnosis is correct but secondary diagnoses are incomplete.**

Figure 12: Criteria for Assessing Dimensional Quality in Reports

---

**Orthopedic Category Classification Prompt**

Classify the orthopedic question into **ONE** category. **Answer ONLY the category name.**

Question: {question} Answer: {answer} Categories:

- **Spine Surgery** - Conditions, injuries, and surgeries related to the spine
- **Foot and Ankle Surgery** - Conditions, injuries, and surgeries related to the foot and ankle
- **Orthopedic Trauma** - Fractures, dislocations, and other acute injuries
- **Hand and Upper Extremity Surgery** - Conditions, injuries, and surgeries related to the hand, wrist, elbow, and shoulder
- **Musculoskeletal Oncology** - Bone and soft tissue tumors
- **Orthopedic Sports Medicine** - Sports-related injuries, arthroscopic surgery
- **Adult Joint Reconstruction** - Arthritis, joint replacement surgery (e.g., hip, knee)

**ANSWER WITH THE EXACT NAME:** "Spine Surgery", "Foot and Ankle Surgery", "Orthopedic Trauma", "Hand and Upper Extremity Surgery", "Musculoskeletal Oncology", "Orthopedic Sports Medicine", "Adult Joint Reconstruction"

Figure 13: Prompt for Orthopedic Category Classification

---

**Spine Category Classification Prompt**

Classify the spine surgery question into **ONE** category. **Answer ONLY the category name.**

Question: {question} Answer: {answer} Categories:

- **General Considerations** - Basic spine anatomy, evaluation, imaging
- **Biomechanics** - Spine mechanics, forces, stability
- **Anatomic Approaches** - Surgical approaches, exposure
- **The Cervical Degenerative Spine** - Cervical disc, stenosis, anterior cervical discectomy and fusion (ACDF)
- **The Thoracic and Lumbar Degenerative Spine** - Lumbar disc, stenosis, fusion
- **Spondylolisthesis** - Spondylolisthesis, pars interarticularis defect
- **Idiopathic Scoliosis** - Adolescent idiopathic scoliosis, curves
- **Adult Spinal Deformity** - Adult scoliosis, sagittal balance
- **Dysplastic and Congenital Deformities** - Dysplastic and congenital deformities
- **Neuromuscular Spine Deformity** - Neuromuscular spinal deformity
- **Kyphosis and Postlaminectomy Deformities** - Kyphosis, post-laminectomy deformities
- **Trauma** - Spine fractures, spinal cord injury
- **Tumor and Osteomyelitis** - Spine tumors, infections
- **Complications** - Surgical complications, internal fixation failure

**ANSWER WITH THE EXACT NAME:** "General Considerations", "Biomechanics", "Anatomic Approaches", "The Cervical Degenerative Spine", "The Thoracic and Lumbar Degenerative Spine", "Spondylolisthesis", "Idiopathic Scoliosis", "Adult Spinal Deformity", "Dysplastic and Congenital Deformities", "Neuromuscular Spine Deformity", "Kyphosis and Postlaminectomy Deformities", "Trauma", "Tumor and Osteomyelitis", "Complications"

Figure 14: Prompt for Spine Category Classification

---

**Generating Medical Q&A for Fine-Tuning Prompt**

"You are a senior clinical medical educator. Please carefully read the provided medical textbook content below and generate **multiple high-quality open-ended question-answer pairs** based on the core knowledge points, all for fine-tuning large language models. Each Q&A should be self-contained and completely independent."

"**Strict Requirements:**"

"1. **Complete Independence**: Each question and answer must constitute a complete knowledge unit that can be understood without any external background materials."

"2. **Prohibited Referential Terms**: Strictly prohibit using terms like "this guide," "the study," "the above materials," "this article," "the report" or any other terms that refer to the original text in questions or answers."

"3. **Clinical Depth Requirements**: Questions should reflect real clinical scenarios, testing deep understanding and clinical thinking rather than simple yes/no questions."

"4. **Open-Ended Design**: Questions should encourage detailed analysis, requiring comprehensive, structured answers that demonstrate clinical reasoning processes."

"5. **Answer Completeness**: Answers must be detailed and comprehensive, including analysis process, reasoning logic, and final conclusions."

"6. **Question Type Diversity**: Should cover multiple dimensions including pathological mechanism explanation, diagnostic thinking analysis, treatment plan design, complication prevention strategies, etc."

"**Question Quantity Requirements:**"

"- Generate appropriate number of questions based on text length"

"- Short text (1-2 paragraphs): Generate 2-3 questions"

"- Medium text (3-5 paragraphs): Generate 3-5 questions"

"- Long text (6+ paragraphs): Generate 5-8 questions"

"**Output Format Requirements:**"

"Strictly follow the XML format below, each textbook page can generate multiple questions:"

"```xml"

"<problem>[Open-ended question 1 stem]</problem>"

"<answer>[Detailed open-ended answer for question 1, including analysis process and conclusions]</answer>"

"<problem>[Open-ended question 2 stem]</problem>"

"<answer>[Detailed open-ended answer for question 2, including analysis process and conclusions]</answer>"

"<problem>[Open-ended question 3 stem]</problem>"

"<answer>[Detailed open-ended answer for question 3, including analysis process and conclusions]</answer>"

"```"

"**Important Notes:**"

"1. Strictly follow the XML format"

"2. Question stems should be clear and specific, encouraging deep thinking, avoiding simple yes/no questions"

"3. Answers should be comprehensive and detailed, including analysis process, reasoning logic, and final conclusions"

"4. Output only XML objects, no additional explanatory text"

"5. Each question should be independent and complete, not dependent on other questions or external materials"

"6. Answers should demonstrate the depth of medical professional knowledge and the logic of clinical thinking"

"**Textbook Content:**content"

"Please generate high-quality medical open-ended question-answer pairs based on the above textbook content."

Figure 15: Prompt for Generating Medical Q&A for Fine-Tuning

---

**Generating Medical MCQs for Fine-Tuning Prompt**

"You are a senior clinical medical educator and examination expert. Please carefully read the provided medical textbook content below and generate **multiple high-quality multiple-choice questions with answers** based on the core knowledge points, all for fine-tuning large language models. Each Q&A should be self-contained and completely independent."
"**Strict Requirements:**"
"1. **Complete Independence**: Each question and options must constitute a complete knowledge unit that can be understood without any external background materials."
"2. **Prohibited Referential Terms**: Strictly prohibit using terms like ̈this guide, ̈the study, ̈the above materials, ̈this article, ̈the report ̈or any other terms that refer to the original text in questions or options."
"3. **Clinical Depth Requirements**: Questions should reflect real clinical scenarios, testing deep understanding and clinical judgment rather than simple memorization."
"4. **Option Design**: Each question must include 4 options (A, B, C, D), with 1 correct answer and 3 high-quality distractors. Distractors should be based on common clinical misconceptions or related concepts."
"5. **Question Type Diversity**: Should cover multiple dimensions including diagnostic reasoning, treatment selection, mechanism explanation, differential diagnosis, complication prevention, etc."
"**Question Quantity Requirements:**"
"- Generate appropriate number of questions based on text length"
"- Short text (1-2 paragraphs): Generate 2-3 questions"
"- Medium text (3-5 paragraphs): Generate 4-6 questions"
"- Long text (6+ paragraphs): Generate 7-10 questions"
"**Output Format Requirements:**"
"Strictly follow the XML format below, each textbook page can generate multiple questions:"
"```xml"
"<problem>[Question 1 stem] A. [Option A] B. [Option B] C. [Option C] D. [Option D]</problem>"
"<answer>[Question 1 correct answer option letter]</answer>"
"<problem>[Question 2 stem] A. [Option A] B. [Option B] C. [Option C] D. [Option D]</problem>"
"<answer>[Question 2 correct answer option letter]</answer>"
"<problem>[Question 3 stem] A. [Option A] B. [Option B] C. [Option C] D. [Option D]</problem>"
"<answer>[Question 3 correct answer option letter]</answer>"
"```"

"**Important Notes:**"
"1. Strictly follow the XML format"
"2. Question stems should be clear and specific, testing deep understanding and clinical judgment"
"3. Options should be reasonably designed, including correct answers and high-quality distractors"

Figure 16: Prompt for Generating Medical MCQs for Fine-Tuning

---

**Generating Context-Localized Multimodal Q&A Prompt**

"You are a senior clinical medical educator. Based on the provided image information, caption, and context, precisely locate the image's position in the context and generate high-quality open-ended questions and answers."

"**Core Task: Multimodal Understanding and Precise Localization Open-Ended Q&A**"

"**Step 1: Multimodal Information Understanding**"

"1. **Image Understanding**: Analyze the specific medical content shown in the image (anatomical structures, pathological manifestations, surgical procedures, imaging features, instrument usage, etc.)"

"2. **Caption Understanding**: Identify figure numbers, positions, operational steps, or key information mentioned in the caption"

"3. **Context Understanding**: Analyze medical knowledge points, operational procedures, and clinical key points in the preceding and following text"

"4. **Position Localization**: Precisely locate the image's specific position and role in the context"

"**Step 2: Precise Position Localization**"

"1. **Caption-Context Matching**:"

" - If the caption contains a figure number (e.g., Figure 12.1), find the corresponding figure reference in the context"

" - If the caption describes operational steps, locate the corresponding operational description in the context"

" - If the caption describes anatomical structures, find related anatomical descriptions in the context"

"2. **Context Position Analysis**:"

" - Analyze preceding text: background information, preparation steps, and related concepts before the image appears"

" - Analyze following text: operational steps, precautions, and clinical significance after the image is shown"

" - Determine the image's specific role in the entire process"

"**Step 3: Generate Open-Ended Q&A Based on Precisely Located Content**"

"Must generate open-ended questions and answers based on precisely located medical knowledge points:"

"- Deeply analyze the relationship between located content and the image"

"- Generate open-ended questions based on precisely located content"

"- Ensure questions are highly relevant to both image content and context"

"- If unable to precisely locate suitable content, skip this question"

"**Step 4: High-Quality Open-Ended Q&A Design**"

"Based on precisely located content, generate open-ended questions and answers with the following characteristics:"

"**Q&A Design Principles:**"

"1. **Multimodal Relevance**: Questions must be relevant to image content, caption information, and context content simultaneously"

"2. **Clinical Orientation**: Questions should be based on real clinical scenarios, testing clinical thinking and decision-making abilities"

"3. **Open-Ended Design**: Encourage deep thinking, avoid yes/no questions, require detailed analysis"

"4. **Position Precision**: Questions should be based on the image's precise location in the context"

"5. **Prohibited Referential Terms**: Strictly prohibit using terms like äccording to the context, this guide, the study, the above materials, this article, the report or any other terms that refer to the original text in questions or answers"

"6. **Answer Design**:"

Figure 17: Prompt for Generating Context-Localized Multimodal Q&A

---

**Generating Context-Localized Multimodal Q&A Prompt**

" - Answers must be based on precisely located content"
" - Cover relevant medical knowledge and clinical considerations"
" - Reflect clinical thinking and decision-making process"
"7. **Question Type Priority**:"
" - Diagnostic analysis questions (in-depth analysis based on imaging findings, clinical symptoms, etc.)"
" - Treatment decision questions (detailed analysis of surgical indications, treatment plan selection, etc.)"
" - Mechanism explanation questions (in-depth explanation of anatomical-physiological relationships, pathological mechanisms, etc.)"
" - Technical operation questions (detailed explanation of surgical steps, instrument usage, etc.)"
" - Risk assessment questions (comprehensive analysis of complication prevention, management strategies, etc.)"
"**Output Format Requirements:**"
"Strictly follow the following XML format, each image can generate multiple different Q&A pairs:"
"```xml"
"<problem><image>
n[First open-ended question stem]</problem>"
"<answer>[First detailed open-ended answer]</answer>"
"<problem><image>
n[Second open-ended question stem]</problem>"
"<answer>[Second answer, directly answering the question]</answer>"
"<problem><image>
n[Third open-ended question stem]</problem>"
"<answer>[Third answer, directly answering the question]</answer>"
"```"
"**Important Notes:**"
"1. Strictly follow the XML format"
"2. Question stems should be clear and specific, encouraging deep thinking, avoiding simple yes/no questions"
"3. Answers should be comprehensive and detailed, including analysis process, reasoning logic, and final conclusions"
"4. If unable to precisely locate relevant content, do not generate questions"
"5. Output only one complete XML object"
"6. Strictly prohibit using referential terms"
"7. **Image Reference Standards**: When referencing images in questions, use general terms like äs shown in the image,̈ ẗhe image displays,̈ ïmaging findingsëtc., strictly prohibit using specific figure numbers (e.g., F̈igure 10.8,̈ F̈igure 12.1,̈ etc.)"
"**Processing Workflow:**"
"1. Analyze the image caption to understand the specific medical content shown"
"2. Precisely locate medical knowledge points in the context related to the caption"
"3. Determine the image's specific position and role in the context"
"4. If precise localization is successful and content is suitable for questions, generate open-ended Q&A based on located content"
"5. If unable to precisely locate or content is not suitable for questions, do not generate questions"
"6. Ensure questions have clinical value and educational significance"
"**Provided Information:**"
"Image Caption: caption"
"Context Information: context"
"Please precisely locate the image's position in the context and generate high-quality medical open-ended questions and answers. If unable to precisely locate suitable content, do not generate questions. Strictly follow the specified XML format."

Figure 18: Prompt for Generating Context-Localized Multimodal Q&A

---

### Generating Context-Localized Multimodal MCQs Prompt

"You are a senior clinical medical educator and examination expert. Your task is to precisely locate the image's position in the context based on the provided image information, caption, and context, then generate high-quality multiple-choice questions."

"**Core Task: Multimodal Understanding and Precise Localization Question Generation**"

"**Step 1: Multimodal Information Understanding**"

"1. **Image Understanding**: Analyze the specific medical content shown in the image (anatomical structures, pathological manifestations, surgical procedures, imaging features, instrument usage, etc.)"

"2. **Caption Understanding**: Identify figure numbers, positions, operational steps, or key information mentioned in the caption"

"3. **Context Understanding**: Analyze medical knowledge points, operational procedures, and clinical key points in the preceding and following text"

"4. **Position Localization**: Precisely locate the image's specific position and role in the context"

"**Step 2: Precise Position Localization**"

"1. **Caption-Context Matching**:"

" - If the caption contains a figure number (e.g., "Figure 12.1"), find the corresponding figure reference in the context"

" - If the caption describes operational steps, locate the corresponding operational description in the context"

" - If the caption describes anatomical structures, find related anatomical descriptions in the context"

"2. **Context Position Analysis**:"

" - Analyze preceding text: background information, preparation steps, and related concepts before the image appears"

" - Analyze following text: operational steps, precautions, and clinical significance after the image is shown"

" - Determine the image's specific role in the entire process"

"**Step 3: Generate Questions Based on Precisely Located Content**"

"Must generate clinical multiple-choice questions based on precisely located medical knowledge points:"

"- Deeply analyze the relationship between located content and the image"

"- Generate multiple-choice questions based on precisely located content"

"- Ensure questions are highly relevant to both image content and context"

"- If unable to precisely locate suitable content, skip this question"

"**Step 4: High-Quality Multiple-Choice Question Design**"

"Based on precisely located content, generate clinical multiple-choice questions with the following characteristics:"

"**Q&A Design Principles:**"

"1. **Multimodal Relevance**: Questions must be relevant to image content, caption information, and context content simultaneously"

"2. **Clinical Orientation**: Questions should be based on real clinical scenarios, testing clinical thinking and decision-making abilities"

"3. **Multiple-Choice Design**: Provide multiple options to test deep understanding and clinical judgment"

"4. **Position Precision**: Questions should be based on the image's precise location in the context"

"5. **Prohibited Referential Terms**: Strictly prohibit using terms like "according to the context," "this guide," "the study," "the above materials," "this article," "the report" or any other terms that refer to the original text in questions or options"

---

Figure 19: Prompt for Generating Context-Localized Multimodal MCQs

---

**Generating Context-Localized Multimodal MCQs Prompt**

"6. **Option Design**:"
" - Correct answer must be based on precisely located content"
" - Distractors should be based on common clinical misconceptions or related but inaccurate concepts"
" - All options should have clinical plausibility"
"7. **Question Type Priority**:"
" - Diagnostic analysis questions (in-depth analysis based on imaging findings, clinical symptoms, etc.)"
" - Treatment decision questions (detailed analysis of surgical indications, treatment plan selection, etc.)"
" - Mechanism explanation questions (in-depth explanation of anatomical-physiological relationships, pathological mechanisms, etc.)"
" - Technical operation questions (detailed explanation of surgical steps, instrument usage, etc.)"
" - Risk assessment questions (comprehensive analysis of complication prevention, management strategies, etc.)"
"**Output Format Requirements:**"
"**Strictly follow the following XML format:**"
""```xml"
"<problem><image>
n[Question stem] A. [Option A] B. [Option B] C. [Option C] D. [Option D]</problem>"
"<answer>[Correct answer option letter]</answer>"
""```"
"**Important Notes:**"
"1. Strictly follow the XML format"
"2. Question stems should be clear and specific, testing deep understanding and clinical judgment"
"3. Options should be reasonably designed, including correct answers and distractors"
"4. If unable to precisely locate relevant content, do not generate questions"
"5. Output only one complete XML object"
"6. Strictly prohibit using referential terms"
"7. **Image Reference Standards**: When referencing images in questions, use general terms like äs shown in the image, ̈the image displays, ̈imaging findingsëtc., strictly prohibit using specific figure numbers (e.g., Ḟigure 10.8, ̈Figure 12.1, ̈etc.)"
"**Processing Workflow:**"
"1. Analyze the image caption to understand the specific medical content shown"
"2. Precisely locate medical knowledge points in the context related to the caption"
"3. Determine the image's specific position and role in the context"
"4. If precise localization is successful and content is suitable for questions, generate multiple-choice questions based on located content"
"5. If unable to precisely locate or content is not suitable for questions, do not generate questions"
"6. Ensure questions have clinical value and educational significance"
"**Provided Information:**"
"Image Caption: caption"
"Context Information: context"
"Please precisely locate the image's position in the context and generate high-quality medical multiple-choice questions. If unable to precisely locate suitable content, do not generate questions. Strictly follow the specified XML format."

Figure 20: Prompt for Generating Context-Localized Multimodal MCQs

