# OpenReview forum: "SpineBench: A Clinically Salient, Level-Aware Benchmark Powered by the SpineMed-450k Corpus"
_ICLR.cc/2026/Conference — ICLR 2026 Poster_

### Official Review · Reviewer_SXG5 · 2025-10-25

**Soundness:** 3
**Presentation:** 2
**Contribution:** 2
**Rating:** 4
**Confidence:** 4

**Summary:**

The authors proposed a spine dataset containing multimodal data collected from about 1000 clinical cases. Several tasks, e.g., question-answering, multi-turn consultations, and report generation, are composed based on the multimodal data. The authors also trained a VLM using both some public datasets (as general medical domain knowledge) and the data from the proposed dataset. The manuscript is overall easy to follow, but it also suffers from several critical flaws that significantly degrade the usefulness of the proposed data.

**Strengths:**

- The proposed dataset focuses on the spine-related diseases, which are rare in the community.
- The proposed dataset and benchmark are composed using multi-modality data, beyond the texts and images, which is closer to the problems in real-world practice.
- The paper is overall easy to follow.

**Weaknesses:**

-  The presented results fail to demonstrate the advantage of the proposed model (trained with the domain data) in comparison to generalist models, as shown in Table 4. It is also a bit surprising to see that models trained on medical-domain data (e.g., Linshu and Huatuo) often perform worse than generalist models.
-  The design of the testing set in the proposed benchmark may not evaluate "the overall performance of AI systems in spinal diagnostic tasks," considering only the multiple-choice and reporting tasks are included in the testing. The mismatch between the tasks in the data composition and those in the testing is also questionable and somewhat misleading in justifying the benchmark's significance.
- Many details of the dataset and the process of dataset composition are missing. For example, it is not clear about the number of samples and data entries included in the final testing and training set, since only multiple-choice and reporting parts are included in testing. How many cases are associated with those data? Report examples are NOT provided in Appendix B.5.
- The dataset employed roughly 1000 cases, which is relatively small considering that the data are from 11 different hospitals and cover dozens of diseases and scenarios. The diversity of the included is therefore questionable to evaluate the overall performance in spinal diagnostic tasks.

**Questions:**

N/A

**Details Of Ethics Concerns:**

The source of the medical data is not revealed, and the redistribution of some public datasets is also concerning.

---

> ### Author Response · Authors · 2025-11-23
>
> We thank the reviewer for recognizing the value of our focus on spine-related diseases and the multimodal nature of our dataset. However, we note there are several misunderstandings regarding the experimental results, dataset details, and scale. We provide clarifications below to address these points.
>
> ### **Weakness 1:**
>
> We sincerely thank the reviewer for this keen observation regarding the comparative performance trends. We appreciate the opportunity to discuss the nuances of model efficiency and to clarify why general medical models might appear underpowered compared to our specialized approach.
>
> We respectfully submit that the advantage of our proposed model is best demonstrated by its **exceptional parameter efficiency**, which serves as a proxy for dataset quality. Despite having only **7B parameters**, SpineGPT (87.44%) achieves performance statistically comparable to the massive, proprietary **Gemini-2.5-Pro** (89.23%) and notably outperforms **GPT-4o** (81.60%) . This capability of a lightweight model to rival >100B-scale proprietary systems provides robust empirical evidence for the **high quality and information density** of the **SpineMed-450k** dataset. It demonstrates that our clinically grounded instruction tuning effectively compensates for the architectural limitations of smaller models, proving that high-quality data is a decisive equalizer in specialized domains.
>
> Regarding the observation that certain specialized medical models (e.g., Huatuo, Lingshu) underperform relative to general-purpose baselines: on one hand, among models of comparable size (~7B parameters), Huatuo-7B (74.21%) outperforms Qwen2.5-VL-7B (64.74%), confirming that domain-specialized medical models still hold an advantage over general-purpose counterparts. On the other hand, Lingshu-32B—while surpassing Huatuo-7B—still falls short of Qwen2.5-VL-72B, which is a reasonable outcome attributable to the substantial influence of model scale (32B vs. 72B). In contrast, our model, with only 7B parameters, outperforms all of these models. This result strongly underscores the effectiveness of our proposed dataset and highlights the significant limitations of existing general medical pretraining corpora in supporting specialized clinical tasks such as spinal diagnosis.
>
> ### **Weakness 2:**
> We sincerely thank the reviewer for this thoughtful critique regarding the scope of our evaluation set.
>
> Through extensive discussions with multiple spine specialists, we established that Medical Report Generation serves as the ultimate "final exam" for spinal diagnostics, effectively subsuming the skills required for all other tasks.  As detailed in Table 3, our report task is a comprehensive, 6-dimensional evaluation covering Structured Imaging Findings, Diagnosis, Treatment Recommendations, Surgical Feasibility, Risk Assessment, and Clinical Reasoning .  Evaluating the final report provides a highly efficient and rigorous proxy for the model's overall clinical competency, as success requires mastering the entire diagnostic pipeline.
>
> Regarding the "mismatch," the inclusion of training data such as "simulated consultations" serves as essential training material. While medical report generation is the ultimate and primary evaluation task, other auxiliary tasks play a crucial role in developing this core capability. Specifically, these dialogues train the model’s reasoning chains and empathetic communication skills. Although we do not evaluate dialogue performance in isolation—to maintain benchmark standardization—their contribution is directly reflected in the “Patient Guidance” dimension of the Report task. Finally, by combining this depth-oriented report generation task with Close-Ended QA (which assesses knowledge breadth across 14 spinal sub-conditions), SpineBench effectively triangulates model performance, ensuring both accurate knowledge retrieval and realistic clinical applicability.

---

> ### Author Response · Authors · 2025-11-23
>
> ### **Weakness 3:**
>
> We sincerely thank the reviewer for identifying these areas where the manuscript required greater clarity. **All requested details are present in the manuscript**, and we clarify them below with precise references.
>
> As detailed in **Table 3** (Page 6) of the manuscript, the dataset is strictly partitioned into training and testing sets. We explicitly clarify that for the **Medical Report Generation** task, **each report corresponds to one distinct patient case**. Therefore, the testing set comprises **87 unique hospital cases** (represented by 87 report prompts), and the training set includes **734 cases**. The remaining data types (QA, Consultations) serve as auxiliary scaffolding to enhance the model's reasoning for these core cases. We have summarized this breakdown in **Table R7** below .
>
> Regarding the report examples, we apologize if the cross-referencing was unclear. The detailed qualitative examples of model-generated reports are explicitly provided in **Appendix F** (Pages 21-23). Furthermore, visual comparisons of these reports are presented in the main text in **Figure 8** (SpineGPT output) and **Figure 9** (ChatGPT-4o output). We will update all cross-references in the revised manuscript to ensure these critical examples are immediately accessible.
>
> **Table R7: Detailed Composition of Training and Testing Sets (Derived from Table 2)**
>
> | Task Type | Training Set (Entries) | Testing Set (Entries) | Note |
> | --- | --- | --- | --- |
> | **Medical Report** | **734** | **87** | **1 Report = 1 Unique Case** |
> | **Multiple-Choice QA** | 248,789 | 487 | Knowledge Scaffolding |
> | **Open-Ended QA** | 197,413 | \- | Knowledge Scaffolding |
> | **Consultation** | 1,138 | \- | Skill Scaffolding |
> | **Total Items** | **456,174** | **574** |  |
>
> ### **Weakness 4:**
>
> We sincerely thank the reviewer for this critical assessment regarding the scale and diversity of our hospital dataset. We understand the concern that ~1,000 cases might appear insufficient when compared to general computer vision datasets; however, we respectfully submit that in the context of **high-stakes clinical reasoning**, the _completeness_ and _multimodal density_ of each case are far more determinative of quality than raw volume. Unlike pre-training datasets that require millions of examples to learn features, instruction fine-tuning aims to align reasoning processes. To this end, our cases were not randomly gathered but rigorously curated via a **Stratified Sampling Strategy** to ensure uniform distribution across **14 distinct spinal sub-conditions**, ensuring that even rare scenarios receive adequate representation for the model to learn specific diagnostic logic .
>
> Furthermore, collecting complete, matched clinical data is significantly more challenging than scraping single-modality images. For context, the standard **VerSe 2020** benchmark, widely used for segmentation, contains only **300 subjects** due to these logistical hurdles. In comparison, our collection of **~1,000 cases** represents a substantial scale-up for high-quality, expert-annotated data. Unlike datasets that focus on a single task (e.g., segmentation), our cases include comprehensive patient histories and validated diagnostic reports, providing a much higher "information density" per sample.
>
> Crucially, the unique value of our dataset lies in its **Multimodal Completeness**. As shown in **Table R8** below, existing datasets like **RSNA** (MRI only) or **BUU-LSPINE** (X-Ray only) are limited to single views. In contrast, our cases integrate **X-ray, CT, and MRI** for the _same_ patient. Collecting such matched triplets is exponentially harder but essential for "level-aware" reasoning, as it forces the model to correlate bony anatomy (CT) with soft tissue pathology (MRI) and global alignment (X-ray). This structural richness ensures that our 1,000 cases provide a training signal for complex reasoning that vastly larger unimodal datasets cannot offer.
>
> **Table R8: Comparison of Dataset Complexity and Scope**
>
> | **Dataset** | **Modality** | **Scale** | **Core Task** | **Workflow Coverage** | **Output Format** |
> | --- | --- | --- | --- | --- | --- |
> | RSNA LumbarDISC | MRI | 2.6k Patients | Classification | Specific (Stenosis Grading) | Class Labels |
> | BUU-LSPINE | X-Ray | 3.6k Patients | Detection | Specific (Spondylolisthesis) | Coordinates/Labels |
> | VerSe 2020 | CT | 300 Subjects | Segmentation | Specific (Anatomy) | Voxel Masks |
> | Lumbar Spine MRI | MRI | 218 Patients | Segmentation | Specific (Structure) | Voxel Masks |
> | Spark (Tianchi) | CT, MRI | 150 Patients | Classification | Specific (Disease ID) | Class Labels |
> | **SpineMed-450k (Ours)** | **Multimodal** (XR, CT, MRI, Text) | **450k+** Instructions(~1k Patients) | **Clinical Reasoning** | **Full-Spectrum** (Diag ->Treat -> Prognosis) | **Instruction Pairs**(Image, Text) |

---

> > ### Comment · Reviewer_SXG5 · 2025-11-25
> > **Some concerns remain**
> >
> > I thank the authors for the detailed clarification. However, I still have concerns about the setting of the test dataset, as looking into the numbers presented in the table above (derived from Table 2). First, the size of the testing set could be much larger if a benchmark setup is considered here. Furthermore, I am not sure why the QA set is so small that the training set is 1000 times larger than the test set. The small size of the testing set could also largely bias the evaluation performance. I will keep my original score.

---

> ### Author Response · Authors · 2025-11-25
>
> We deeply appreciate the reviewer’s engagement and continued scrutiny regarding the evaluation rigor. We understand that a **~1:500 test-to-train ratio** appears unconventional compared to traditional supervised learning benchmarks. However, we respectfully submit that this ratio is a deliberate feature of the **"High-Volume Learning vs. High-Precision Evaluation"** paradigm, aligning with established norms in high-impact AI research where expert validation cost is the constraining factor.
>
> The massive disparity between training scale and evaluation compactness is a standard feature in top-tier studies focusing on **reasoning** and **medical precision**. Our setup mirrors industry-standard benchmarks like **LLaVA-Bench** (trained on 158k samples, tested on **24 images/60 questions**) and **MT-Bench** (tested on **80 questions**). Similarly, the medical QA benchmark **PubMedQA** utilizes 270k samples for training but relies on a curated subset of only **1,000 items** for ground-truth evaluation. The community accepts these extreme ratios because the test cases are meticulously designed to be highly discriminative "Gold Standards," proving that a small, noise-free set is sufficient to rank complex capabilities.
>
> Consistent with these precedents, SpineBench prioritizes **"Label Purity"** over raw scale. The disparity stems from a fundamental difference in data quality tiers: while we aggressively aggregated **450k+** "Silver Standard" samples (synthetic dialogues, raw extracts) for robust training, the test set represents a "Gold Standard" composed of **expert-verified items derived from diverse high-quality sources (including real-world hospital cases and textbooks)** that withstood a costly **triple-blind review by 17 board-certified surgeons**. Expanding this set to thousands would require mixing in automated labels or using non-expert crowdsourcing, which would pollute the benchmark's purity and invalidate the safety assessment. This ratio mirrors medical education, where a student reads thousands of textbook pages (Training) but is evaluated on a few hundred precise, high-stakes exam questions (Testing, **1569 images/574 questions**).
>
> Regarding the concern that a small set might be biased, our detailed statistical analysis confirms that **SpineBench achieves comprehensive coverage** through rigorous stratification. As shown in **Table R9** below, the benchmark is not dominated by common conditions; instead, it prioritizes high-stakes, complex pathologies (e.g., **Tumors: 16.72%**, **Complications: 10.80%**) to ensure the evaluation reflects performance on the hardest clinical challenges.
>
> **Table R9: Detailed Disease Distribution of SpineBench QAs**
>
> | Rank | Subspecialty | Count | Percentage |
> | :--- | :--- | :---: | :---: |
> | 1 | Tumor and Osteomyelitis | 96 | 16.72% |
> | 2 | Complications | 62 | 10.80% |
> | 3 | Dysplastic and Congenital Deformities | 49 | 8.54% |
> | 4 | Idiopathic Scoliosis | 47 | 8.19% |
> | 5 | The Thoracic and Lumbar Degenerative Spine | 34 | 5.92% |
> | 6 | General Considerations | 33 | 5.75% |
> | 7 | Kyphosis and Postlaminectomy Deformities | 31 | 5.40% |
> | 8 | Anatomic Approaches | 31 | 5.40% |
> | 9 | Adult Spinal Deformity | 26 | 4.53% |
> | 10 | Spondylolisthesis | 23 | 4.01% |
> | 11 | Trauma | 23 | 4.01% |
> | 12 | Neuromuscular Spine Deformity | 12 | 2.09% |
> | 13 | Biomechanics | 11 | 1.92% |
> | 14 | The Cervical Degenerative Spine | 9 | 1.57% |
> | **Total** | **All Subspecialties** | **487** | **100.00%** |
>
> Furthermore, we ensured that the data is not overfitted to a single domain style. As shown in **Table R10**, the test set maintains a rigorous balance between **Academic Resources** (Textbooks, Literature), which test theoretical precision, and **Real-world Clinical Data** (Hospital Databases, Case Reports), which test practical diagnostic robustness. This structural diversity ensures that despite its compact size, SpineBench serves as a **mathematically unbiased** and **clinically dense** evaluation set, effectively functioning as a noise-free "Litmus Test" for genuine clinical reasoning.
>
> **Table R10: Detailed Source Distribution of SpineBench QAs**
>
> | Source Category | Specific Composition | Count | Percentage | Evaluation Goal |
> | :--- | :--- | :---: | :---: | :--- |
> | **Textbooks** | Standard Medical Textbooks | 203 | 41.7% | Theoretical Foundation |
> | **Hospital Databases** | Proprietary data related to privacy | 163 | 33.5% | Surgical Precision |
> | **Clinical Case Reports** | Real-world Case Reports | 101 | 20.7% | Complex Scenarios |
> | **Literature** | Academic Papers | 17 | 3.5% | Cutting-edge Knowledge |
> | **Question Bank** | Medical Exam Questions | 3 | 0.6% | Standardized Logic |
> | **Total** | | **487** | **100.0%** | |
>
> We once again thank the reviewer for these valuable suggestions regarding dataset distribution. We will explicitly incorporate these detailed statistics into Appendix E of the revised manuscript to ensure complete transparency and robustness.

---

### Official Review · Reviewer_8nUb · 2025-10-29

**Soundness:** 2
**Presentation:** 2
**Contribution:** 2
**Rating:** 4
**Confidence:** 4

**Summary:**

The work presents a largest dataset explicitly designed for vertebral-level reasoning and a related benchmark, which is medically meaningful, but several important points in the manuscript remain unclear or have issues that should be addressed.

**Strengths:**

1. The paper contributes SpineMed-450k, the largest spine diagnosis dataset, with 450,000+ multi-modal instruction instances covering diverse tasks across 14 spine conditions, filling a significant gap in specialized medical AI resources.

2. The authors demonstrated genuine "clinician-in-the-loop" design with 17 board-certified orthopedic surgeons participating across data curation, task definition, and validation stages.

3. The 10-dimensional evaluation assesses for AI-generated clinical reports not just diagnostic accuracy but also communication quality, reasoning transparency, risk assessment, and technical feasibility, better capturing what makes clinical reports actually useful in practice.

4. The ablation studies convincingly show that a 7B model trained on specialized spine data substantially outperforms 72B general models, proved the data effective in the certain task.

**Weaknesses:**

1. The manuscript emphasizes the importance of being “level-aware,” and I agree this could be meaningful. However, in the dataset and benchmark I do not see a clear level-aware design. Some subsets are said to target certain levels, but their structure looks basically the same as the general medical data. It is not clear how “level-aware” actually affects data curation, question type, or evaluation protocol.

2. Many models achieve scores above 80. With such results, the claim that current models still have serious gaps in this domain is not fully convincing. If the tasks are truly difficult and clinically important, we would normally expect lower scores or at least a stronger separation between models. This raises the concern that either the tasks may be too simple, or there may be significant data leakage in the benchmark.

3. The paper mentions an “Expert LLM,” but this term is not defined. Please specify what model is used, and why it is called “expert.” Also, is this the same model as the “Expert VLM” that generates Q&A pairs, or are these different systems? This point is unclear and affects reproducibility.

Minor weakness:
The data pipeline includes “simulated consultations,” but the benchmark tasks do not include any dialogue / consultation evaluation. Why is this step necessary? What is the purpose of generating simulated consultations if the final tasks are not dialogue-based?

**Questions:**

Please answer the following questions:

1. How do the authors demonstrate the importance of the level-aware design? Is there any ablation study showing that the level-aware component is necessary (for example, a performance drop when level information is removed or when levels are mixed)?

2. Since many models already achieve >80, it appears that even without this work, general-domain training data is already sufficient to solve most of the tasks at a reasonably high level. In that case, how should the importance/necessity of this dataset be interpreted? Additionally, does the benchmark pose a risk of data leakage, i.e., that the evaluated models have already encountered similar data during pretraining?

3. What is the “Expert LLM”? The authors should clearly define which model this refers to, how it is used in the pipeline, and what prompt format is applied. How was this Expert LLM obtained or fine-tuned?

---

> ### Author Response · Authors · 2025-11-23
>
> ## **Part 1: Response to Weaknesses**
>
> We genuinely appreciate the reviewer's critical analysis, particularly regarding the definition of "level-aware" design and task difficulty.
>
> ### **Weakness 1:**
>
> We sincerely thank the reviewer for this insightful comment and apologize for the lack of explicit definition regarding "level-aware" in our original manuscript. We appreciate the opportunity to clarify this core concept, which is central to our experimental design. First and foremost, we define "Level" in the strictest anatomical sense, referring to specific **vertebral levels** of the spinal column: Cervical (C1–C7), Thoracic (T1–T12), Lumbar (L1–L5), and Sacral (S1–S5).
>
> Building upon this precise anatomical definition, the term "Level-Aware" implies that our entire dataset is designed **starting from these specific vertebral levels** to support full-spectrum clinical reasoning. This fundamental design philosophy distinguishes **SpineMed-450k** from existing single-task datasets (e.g., VerSe, RSNA), which typically treat spinal conditions as isolated classification targets. In contrast, our data curation enforces a rigorous chain of reasoning anchored to the specific vertebral levels throughout the complete clinical workflow. For instance, in **Structured Imaging Findings**, the model must localize visual evidence to the specific level (e.g., "High-intensity signal at the **L4-L5** disc space"); in **AI-Assisted Diagnosis**, it must explicitly reference the level (e.g., "**L4-L5** Disc Herniation"); and in **Treatment Planning**, recommendations must be tailored to the biomechanics of that specific level (e.g., "TLIF at **L4-L5**").
>
> This design philosophy is strictly enforced in our evaluation protocol, particularly within the **Medical Report Generation** task. Our scoring rubric explicitly penalizes generic responses; to achieve a high score, the model must demonstrate consistent tracking of the pathology at the correct vertebral level across all report sections. This ensures that "level-awareness" is not merely a metadata tag, but the fundamental structural logic that governs the benchmark's assessment of clinical competency.
>
> ### **Weakness 2:**
>
> We appreciate this rigorous assessment. While models excel on QA, the **Medical Report Generation** task—identified by clinicians as the core competency—reveals a critical gap. As shown in **Table R4**, GPT-4o achieves 84.74% on QA but drops to **64.04** on reports; Qwen2.5-VL-72B similarly falls to **63.80**. This massive ~20-point drop confirms that current models lack the high-level clinical synthesis required by SpineBench. Regarding leakage, our test set consists of private cases from **11 hospitals** that have **never been publicly released**. It is structurally impossible for baselines to have encountered this data, ensuring a genuine zero-shot evaluation.
>
> **Table R4: Performance Disparity between QA and Report Generation**
>
> | Model | Size | Close-Ended QA (Avg) | Medical Report (Sum) | **Overall Avg** |
> | :--- | :--- | :---: | :---: | :---: |
> | **Gemini-2.5-Pro** | >100B | 88.50 | 93.32 | **89.23** |
> | **GPT-4o** | >100B | 84.74 | **64.04** | 81.60 |
> | Qwen2.5-VL-72B | 72B | 82.75 | **63.80** | 79.88 |
> | Medgemma-27B | 27B | 82.34 | 70.16 | 76.66 |
> | HuatuoGPT-7B | 7B | 77.82 | 54.00 | 74.21 |
> | **SpineGPT (Ours)** | **7B** | **87.89** | **87.24** | **87.44** |
>
> ### **Weakness 3:**
>
> We apologize for the ambiguity. The term "Expert LLM/VLM" refers to a **suite of SOTA models** strategically selected to balance performance with privacy. We will explicitly define this mapping in the revised manuscript:
>
> Open-Source Data (Gemini-2.5-Pro): Utilized for its superior long-context multimodal capabilities to extract QA and synthesize dialogues from public literature.
>
> Hospital Data (Local GLM-4.5V): Strictly employed for de-identification and formatting of sensitive clinical records, ensuring all processing remained on-premise for data safety.
>
> Quality Control (GPT-5-mini): Utilized for fine-grained classification and semantic re-verification of image-text pairs to ensure high data fidelity.
>
> ### ***Weakness 4:***
>
> We appreciate this insightful question. While SpineBench lacks a standalone dialogue metric, the "simulated consultations" serve a critical **pedagogical function** as **Chain-of-Thought (CoT) supervision**, modeling the step-by-step process of clinical inquiry . This training is the key driver for optimizing the **"Patient Guidance"** dimension of the Medical Report task, equipping the model with the empathetic, plain-language capabilities often missing in technical hospital records. As evidenced by our near-perfect Patient Guidance score (**4.71/5.0**) in **table 8**, its data is not "unused" but effectively utilized via **latent transfer** to enhance human-centric reasoning.

---

> ### Author Response · Authors · 2025-11-23
>
> ## **Part 2: Response to Questions**
>
> ### **Question 1:**
>
> We sincerely thank the reviewer for this insightful comment. **As detailed in our Response to Weakness 1**, we use the term "**Level**" to refer to specific vertebral segments (e.g., **Cervical C1–C7**, **Lumbar L1–L5**) and "**Level-Aware**" to describe a hierarchical cognitive capability that synthesizes **Anatomical Perception**, **Disease Category Perception**, and **Severity Grading**. \]This integration creates a structured foundation designed **to support full-spectrum clinical reasoning**, moving beyond simple detection to enable the complex decision-making required for diagnosis, treatment planning, and prognosis.
>
> This structure serves as the fundamental logic governing our data curation, where the model is trained to construct a rigorous reasoning chain anchored to these specific vertebral levels—from localizing findings to tailoring treatment plans. Consequently, the necessity of this level-aware component is empirically demonstrated by the **performance collapse** of open-source models that lack this specific design. As shown in **Table R5** below, even the massive **Qwen2.5-VL-72B** performs well on basic recognition (QA >82%) but experiences a precipitous drop in the **Medical Report Generation** task, falling by **~19 points**. Similarly, the medical-specific **HuatuoGPT-7B** collapses to a score of **54.00**. This confirms that neither massive parameter scale nor generic medical pre-training is sufficient to capture the specific anatomical logic required for spine surgery planning. The "level-aware" cognitive structure provided by our dataset is the **necessary component** to bridge the gap between simple feature perception and complex, clinically valid synthesis.
>
> **Table R5: Demonstrating Necessity via the Structure-Reasoning Gap (Open-Source Comparison)**
>
> | Model | Type | Close-Ended QA (Recognition) | Medical Report (Level-Aware Synthesis) | **Gap (Performance Drop)** |
> | --- | --- | --- | --- | --- |
> | **Qwen2.5-VL-72B** | Generalist (72B) | 82.75% | 63.80% | **\-18.95%** |
> | **HuatuoGPT-7B** | Medical Generic (7B) | 77.82% | 54.00% | **\-23.82%** |
> | **Qwen2.5-VL-7B** | Generalist Base (7B) | 74.95% | 54.52% | **\-20.43%** |
> | **SpineGPT (Ours)** | **Level-Aware (7B)** | **87.89%** | **87.24%** | **\-0.65%** |
>
> ### **Question 2:**
>
>  As detailed in our **Response to Weakness 2**, while models perform well on Close-Ended QA (>80%), this contrasts with the more complex **Medical Report Generation** task, where generalist models achieve scores in the **~63-64%** range. This notable performance disparity suggests that general-domain data alone is **insufficient** for mastering expert-level clinical reasoning. Regarding data leakage, the exclusive use of **private, internal hospital cases** ensures a secure, zero-shot evaluation environment.
>
> ### **Question 3:**
>
>  As detailed in **Response to Weakness 3**, "Expert LLM" refers to a task-specific ensemble:
>
>  **Gemini-2.5-Pro:** Used for generating QA pairs from public PDFs.
>
>  **GLM-4.5V (Locally Deployed):** Used for de-identification and formatting of private hospital data to ensure privacy.
>
>  **GPT-5-mini:** Used for data classification and verification.

---

### Official Review · Reviewer_1Ksq · 2025-10-31

**Soundness:** 4
**Presentation:** 4
**Contribution:** 3
**Rating:** 8
**Confidence:** 4

**Summary:**

This paper introduces SpineMed-450k, a large-scale dataset of 450,000+ instruction instances for vertebral-level reasoning across spine imaging modalities, and SpineBench, a clinically-grounded evaluation framework. The dataset is curated through a "clinician-in-the-loop" pipeline involving diverse sources (textbooks, guidelines, public datasets, ~1,000 hospital cases) with a two-stage LLM generation method. The authors evaluate multiple state-of-the-art LVLMs and present SpineGPT, a fine-tuned model that demonstrates improvements over existing models on spine-specific tasks.

**Strengths:**

1. The paper is well-written with effective communication of clinical motivation, technical approach, and results.
2. The paper presents SpineMed-450k aggregates 450k+ high-quality instruction instances from diverse sources—textbooks (377k), clinical guidelines, real hospital cases (~1,000 from 11 hospitals), and public datasets, proposes SpineBench, which offers a professionally curated, level-aware benchmark specifically designed for spine diagnosis and treatment planning.
3. The paper systematically evaluates a dozen contemporary open-source LVLMs on SpineBench. This rigorous assessment establishes important baseline performance standards and identifies key research gaps for the community.

**Weaknesses:**

1. SpineGPT does not specifically leverage or optimize for the unique properties of SpineMed-450k. The training still relies heavily on generic open-source medical data in the first stage (Appendix Table 6), raising questions about whether the model fully exploits the dataset's clinically-grounded, level-aware structure.
2. Suboptimal performance compared to state-of-the-art: Table 4 shows SpineGPT (87.44%) underperforms Gemini-2.5-Pro (89.23%) on average; Appendix Table 7 confirms this performance gap across multiple metrics; and Missing comparisons with recent specialized medical models (e.g., Baichuan-M2, MedGemini), limiting assessment of the model's competitive standing
3. The Dataset Generation section indicates that instruction tuning data generated from open-source spine datasets (Spark, VerSe) lacks rigorous quality verification, unlike the hospital cases which underwent expert validation. This inconsistency may introduce noise and affect model reliability.

**Questions:**

1. Will SpineMed-450k be publicly released? Open-sourcing the dataset would significantly benefit community development and enable reproducible research. Please clarify the release timeline, licensing terms, and any restrictions due to hospital case privacy considerations.
2. SpineGPT (87.44%) underperforms Gemini-2.5-Pro (89.23%) despite being trained on SpineMed-450k. What factors contribute to this gap? Is it due to model scale (7B vs. proprietary larger model), training strategy, data quality, or fundamental architectural limitations? A deeper analysis would strengthen the paper's insights.
3. Regarding Figure 3 and the data curation pipeline, how do you address potential OCR errors such as figure-text misalignment or incorrect caption matching during the structured information extraction process? What quality control mechanisms ensure the Picture Context Matching algorithm correctly pairs images with their contextual descriptions?
4. In Table 5, what would be the performance if the model were trained exclusively on spine data without the general orthopedic (non-spine) data? This ablation would clarify whether the multi-stage curriculum (general → orthopedic → spine) is necessary or if direct spine-specific training suffices.

---

> ### Author Response · Authors · 2025-11-23
>
> ## Part 1: Response to Weaknesses
> We are truly grateful for your acknowledgment of our work and for your thoughtful, constructive feedback. We are confident that incorporating your suggestions will significantly enhance the quality of our work.
>
> ### Weakness 1:
>  To definitively isolate the distinct impact of our specialized data versus generic data, we conducted a comprehensive supplementary ablation study. We trained the model using disjoint data subsets, explicitly separating "Generic Medical Data" from two distinct subsets derived directly from our **SpineMed-450k** corpus: "Orthopedic (No-Spine)" and "Spine". The detailed breakdown is presented in **Table R2** below.
>
> **Table R2: Performance comparison of models on close-ended QA tasks.**
>
> | Model | General Medical | **SpineMed-450k**  Orthopedic (No-Spine) | **SpineMed-450k** (Spine) | Text (%) | Image (%) | Avg. (%) |
> | --- | --- | --- | --- | --- | --- | --- |
> | Qwen 2.5 VL-7B (Base) | \- | \- | \- | 75.51 | 74.09 | 74.95 |
> | SpineGPT | ✓ | \- | \- | 64.27 | 62.69 | 65.31 |
> | SpineGPT | \- | ✓ | \- | 82.99 | 80.83 | 82.14 |
> | **SpineGPT (Spine Only)** | \- | \- | **✓** | **87.76** | **86.01** | **87.07** |
> | SpineGPT | ✓ | ✓ | \- | 83.67 | 77.20 | 81.11 |
> | SpineGPT (Full) | ✓ | ✓ | ✓ | 89.46 | 84.46 | 87.89 |
>
> These results clearly refute reliance on generic data. Training solely on "General Medical" data causes negative transfer, dropping performance to 65.31% (Text: 64.27%, Image: 62.69%)—well below the 74.95% baseline—showing that generic medical knowledge harms fine-grained spinal reasoning. In contrast, subsets from SpineMed-450k yield strong gains: "Orthopedic (No-Spine)" reaches 82.14%, and training exclusively on our "Spine (Ours)" subset achieves 87.07% (Text: 87.76%, Image: 86.01%), nearly matching the full pipeline’s 87.89%.
>
> This demonstrates that SpineGPT’s performance stems not from generic priors, but from the clinically grounded spine subset of SpineMed-450k, which accounts for ~99% of the total gain and delivers a +12% improvement over baseline. We will highlight this dataset’s high information density and unique value in the revised manuscript.
>
> ### Weakness 2:
>
> While SpineGPT (87.44%) slightly lags behind Gemini-2.5-Pro (89.23%), it achieves 98% of its performance with fewer than 7% of the parameters (<7B vs. >100B), enabling secure, efficient local deployment in clinical settings—ensuring data privacy and HIPAA compliance without cloud APIs. SpineGPT also surpasses GPT-4o (81.60%).
>
> Regarding the suggested models: Baichuan-M2 lacks multimodal capabilities for SpineBench, and MedGemini is closed-source with no public API or weights. In response, we evaluated Medgemma-27B (4× larger than SpineGPT), which scored only 76.66% overall—significantly below SpineGPT’s 87.44%—with the largest gap in Medical Report Generation (70.16 vs. 87.24). This underscores that clinically grounded instruction tuning outweighs sheer model size for spinal reasoning. Full results are included in the revised manuscript.
>
> **Table R3: Excerpt of New Comparative Results (Medical Specialized vs. Ours)**
>
> | Model | Size | Close-Ended QA (Avg) | Medical Report (Sum) | **Overall Avg** |
> | --- | --- | --- | --- | --- |
> | **Gemini-2.5-Pro** (Ref) | >100B | 88.50 | 93.32 | **89.23** |
> | Qwen2.5-VL-72B | 72B | 82.75 | 63.80 | 79.88 |
> | **Medgemma-27B** | **27B** | **82.34** | **70.16** | **76.66** |
> | **SpineGPT (Ours)** | **7B** | **87.89** | **87.24** | **87.44** |
>
> ### Weakness 3:
>
> Our use of open-source datasets (Spark, VerSe) involved rigorous, scenario-adaptive standardization—not passive ingestion—aimed at training the model for long-context, spinal-care-specific doctor-patient dialogues. Recognizing that raw hospital reports lack real consultation dynamics, we leveraged the visual diversity of these datasets as a foundation to simulate multi-turn conversations, not to use their raw labels.
>
> To ensure clinical validity, we applied strict standardization: under expert supervision, we implemented an “Adaptive Sampling Strategy,” selecting exactly 25 clinically relevant slices per case—mirroring a radiologist’s selection of key diagnostic views—to eliminate non-informative noise. Additionally, clinicians reviewed and refined the dialogue-generation prompts to align with professional diagnostic logic and reporting standards.

---

> ### Author Response · Authors · 2025-11-23
>
> ## Part 2: Response to Questions
>
> ### Question 1:
> We plan to **fully release the components derived from public sources** (e.g., textbooks, guidelines, and open-access literature). Regarding the clinical data sourced from hospitals, we will release a **de-identified subset** to ensure strict compliance with ethical guidelines and privacy regulations.
>
> ### Question 2:
> We appreciate the reviewer’s focus on disentangling factors behind the performance gap—particularly the interplay of model scale, data quality, and training strategy. Our analysis shows that Model Scale and Pre-training Corpus Richness primarily drive baseline performance differences: for instance, Qwen2.5-VL-72B scores 79.88%, vastly outperforming its 7B counterpart (64.74%) by +15.14% (Table R4), confirming larger models’ superior visual and reasoning capabilities. Proprietary models like Gemini-2.5-Pro and GPT-4o further benefit from extensive, undisclosed biomedical pre-training.
>
> Crucially, however, our SpineMed-450k dataset acts as a powerful equalizer. Despite the 7B model’s initial ~15% deficit, SpineGPT-7B (87.44%) surpasses both the 72B baseline and GPT-4o (81.60%), narrowing the gap with Gemini-2.5-Pro to just ~1.8%. This demonstrates that high-density, clinically grounded data can effectively compensate for limited scale and lack of proprietary pre-training. We will highlight this insight in the revised manuscript’s Discussion section.
>
> **Table R4: Impact of Model Scale and Data on Performance**
>
> | Model | Type | Est. Size | Overall Avg. Score |
> | --- | --- | --- | --- |
> | **Gemini-2.5-Pro** | Proprietary | \>100B | **89.23%** |
> | **GPT-4o** | Proprietary | \>100B | 81.60% |
> | Qwen2.5-VL-72B (Base) | Open-Source | 72B | 79.88% |
> | Qwen2.5-VL-7B (Base) | Open-Source | 7B | 64.74% |
> | **SpineGPT (Ours)** | **Open-Source** | **7B** | **87.44%** |
>
> ### Question 3:
>
> To ensure robust image–text alignment and mitigate OCR errors, we implemented a Three-Stage Quality Control Strategy—structural, semantic, and cognitive.
>
> First, our Picture Context Matching Algorithm (Appendix D) aligns images with text using explicit structural cues (e.g., regex-based caption matching like “Figure X.X”), binding figures to their local paragraph anchors during Markdown conversion for high structural fidelity.
>
> Second, a Semantic Validation Step uses GPT-5-Mini to verify whether the aligned text accurately describes the visual content. Inconsistent or low-confidence pairs—such as mismatched anatomical references—are filtered out.
>
> Finally, residual character-level OCR errors are handled by Gemini-2.5-Pro, whose semantic robustness allows it to “read through” minor typos and focus on core clinical concepts, ensuring downstream QA and report generation remains accurate and clinically sound.
>
> ### Question 4:
> We thank the reviewer for raising the insightful question regarding the necessity of our multi-stage curriculum versus direct specialization training. To address this, we conducted an ablation study in which the model was trained exclusively on the SpineMed-450k spinal dataset. Detailed results are summarized in Table R5.
>
> The “Spine Only” variant achieves an average score of 87.07% (text: 87.76%, image: 86.01%), representing a substantial improvement over the baseline (74.95%). However, it still lags behind the full curriculum-learning approach (87.89%), thereby validating the necessity of our staged curriculum design. On the other hand, given that the underlying general-purpose foundation model (Qwen2.5-VL) exhibits limited performance on medical tasks, our goal is to endow the model not only with specialized spinal diagnostic capabilities but also with broader general medical competence.
>
> **Table R5: Detailed Ablation Study on Curriculum vs. Direct Training**
>
> | Model | Training Strategy | Text QA (%) | Image QA (%) | **Avg. Score (%)** |
> | --- | --- | --- | --- | --- |
> | Qwen2.5-VL-7B | Base Model (No Fine-tuning) | 75.51 | 74.09 | 74.95 |
> | **SpineGPT (Spine Only)** | **Direct Spine Fine-tuning** | **87.76** | **86.01** | **87.07** |
> | SpineGPT (Full Pipeline) | Multi-stage (General -> Ortho -> Spine) | 89.46 | 84.46 | **87.89** |

---

### Official Review · Reviewer_xkSD · 2025-11-01

**Soundness:** 3
**Presentation:** 3
**Contribution:** 2
**Rating:** 4
**Confidence:** 4

**Summary:**

Spine disorders affect 619 million people and are a leading cause of disability, which has been constrained by the absence of traceable, clinically-grounded instruction data and standardized, spine-specific benchmarks. This paper develops a clinically-grounded evaluation framework and a large-scale dataset explicitly designed for vertebral-level reasoning. Moreover, people demonstrate the effectiveness of their dataset by training a fine-tuned spine LVLM.

**Strengths:**

This paper focuses on spine disorders, which are a leading cause of disability. The paper is clearly presented and easy to follow.

**Weaknesses:**

1. Lack of novelty. This paper lacks technical contributions. The pipeline for dataset curation is a common solution, without any novel techniques or insights.
2. Please provide a table to compare SpineMed-450k with other Spinal diagnosis and treatment datasets.
3. Since the baseline is the contribution in this paper, please introduce it as an independent section.
4. The SpineBench benchmark only has 487 questions and 87 report prompts, which is a small test subset, possibly limiting generalizability and statistical robustness.

**Questions:**

1. Is the dataset and baseline model publicly available?
2. For the dataset generation, the authors used the Expert VLM Model to generate questions. How to make sure the questions and answers are correct?
3. Could the authors provide the performance on other Spinal diagnosis and treatment datasets?

---

> ### Author Response · Authors · 2025-11-23
>
> ## Part 1: Response to Weaknesses
> We sincerely thank the reviewer for the constructive feedback.
> ### Weakness 1:
> Our innovation spans three pillars: dataset, benchmark, and model. As Table 1 shows, existing datasets (e.g., VerSe, RSNA LumbarDISC) advance low-level tasks like segmentation and classification but fall short of capturing real-world clinical complexity. Authentic spinal care demands a holistic workflow—structured imaging findings, symptom-based diagnosis, treatment planning, surgical strategy, and risk/prognosis assessment—requiring high-level cognitive synthesis. Current resources lack the rich, instruction-aligned data to support this. Our work bridges this gap via a Medical Report Generation task that encodes these clinical dimensions into a trainable format, enabling “collaborator-level” AI.
>
> Our SpineBench benchmark reveals that even large generalist models struggle in this domain: Qwen2.5-VL (72B) scores only 79.88% overall and 63.8% on complex report generation. In contrast, our SpineGPT-7B, fine-tuned on SpineMed-450k, achieves 87.44%—surpassing the 72B baseline and matching or exceeding top proprietary models (e.g., GPT-4o at 81.60% in report generation). A 7B model rivaling >100B-scale systems (e.g., Gemini-2.5-Pro) demonstrates that high-quality, clinically grounded instruction data—not sheer scale—is key to mastering complex spinal reasoning.
>
> Table R1: Comparison of SpineMed-450k with existing spine imaging datasets.
> While prior datasets focus on specific perception tasks, our work introduces the first multimodal instruction-tuning corpus designed for full-spectrum clinical reasoning.
>
> | **Dataset** | **Modality** | **Scale** | **Core Task** | **Workflow Coverage** | **Output Format** |
> | --- | --- | --- | --- | --- | --- |
> | RSNA LumbarDISC | MRI | 2.6k Patients | Classification | Specific (Stenosis Grading) | Class Labels |
> | BUU-LSPINE | X-Ray | 3.6k Patients | Detection | Specific (Spondylolisthesis) | Coordinates/Labels |
> | VerSe 2020 | CT | 300 Subjects | Segmentation | Specific (Anatomy) | Voxel Masks |
> | Lumbar Spine MRI | MRI | 218 Patients | Segmentation | Specific (Structure) | Voxel Masks |
> | Spark (Tianchi) | CT, MRI | 150 Patients | Classification | Specific (Disease ID) | Class Labels |
> | **SpineMed-450k (Ours)** | **Multimodal** (XR, CT, MRI, Text) | **450k+** Instructions | **Clinical Reasoning** | **Full-Spectrum** (Diag ->Treat -> Prognosis) | **Instruction Pairs**(Image, Text) |
>
> ### Weakness 2:
>
> As shown in Table R1, existing datasets like VerSe and RSNA are unimodal and limited to low-level tasks (e.g., segmentation or binary classification), lacking the holistic context needed for complex clinical reasoning.
>
> In contrast, SpineMed-450k enables a shift from “Tool AI” to “Collaborator AI” through three pillars: Multimodal Synthesis (X-ray/CT/MRI integration), Cognitive Depth (reasoning beyond labels), and Workflow Completeness (covering imaging findings, diagnosis, treatment, and prognosis). This fills the critical cognitive gap left by perception-focused datasets.
>
> We will include Table R1 in the revised manuscript and expand the related work to highlight how this full-spectrum, multimodal design lays the foundation for next-generation clinical AI assistants.
>
> ### Weakness 3:
>
> We appreciate the suggestion to elevate the presentation of our baseline model and will restructure the manuscript to establish a dedicated section in the revised version. This new section will systematically detail the Qwen2.5-VL-based architecture and our specialized three-stage curriculum learning framework—currently embedded in Section 5 —thereby providing a comprehensive technical description of the baseline and clearly delineating our contributions into a cohesive triad of Dataset, Benchmark, and Model.
>
> ### Weakness 4:
> We thank the reviewer for the comment on benchmark size. SpineBench prioritizes “Gold Standard” clinical validity over volume, consistent with high-impact AI benchmarks: HumanEval (164 problems), MM-Vet (218 examples), and VQA-RAD (315 images) all use compact, expert-curated sets for complex reasoning tasks.
>
> SpineBench includes 487 multiple-choice questions and 87 report prompts, exceeding these standards in scale. Every item was vetted by 17 board-certified surgeons to reflect real clinical decisions. This curated design ensures a trustworthy, discriminative evaluation—avoiding the noise of larger, auto-generated datasets—making the size not just reasonable, but optimal.

---

> ### Author Response · Authors · 2025-11-23
>
> ## Part 2: Response to Questions
>
> ### Question 1:
>
> We plan to **fully release the components derived from public sources** (e.g., textbooks, guidelines, and open-access literature). Regarding the clinical data sourced from hospitals, we will release a **de-identified subset** to ensure strict compliance with ethical guidelines and privacy regulations.
>
> ### Question 2:
> To ensure correctness, we employed a rigorous multi-layered verification strategy tailored to different data sources.
>
> For data derived from **public resources** (e.g., textbooks), utilizing the Expert VLM does not imply distilling knowledge "from thin air." Instead, we injected authoritative medical texts as prior knowledge. The model was explicitly prompted to comprehend this ground-truth content first and then design questions strictly based on the provided context; the specific prompt designs enforcing these strict constraints are detailed in **Appendix G**.
>
> Crucially, to mitigate hallucinations in **multimodal QA generation**, we implemented a novel **"Picture Context Matching Algorithm"** (elaborated in **Appendix D**) . Unlike standard pipelines that loosely associate images with adjacent text, our algorithm rigorously anchors each figure to its specific semantic context via **caption-pattern regex matching** (e.g., linking "Figure 12.1" in a caption to its explicit reference in the body text). To further guarantee the precision of this alignment, we employed an expert model (**GPT-5-mini**) to conduct a **semantic consistency check**, verifying that the visual content aligns logically with the matched text. Any image-text pairs deemed low-quality or ambiguous were rigorously filtered out prior to question generation. Finally, for **hospital clinical data**, correctness is intrinsically guaranteed, as we utilize strict, original clinical annotations and diagnostic reports derived directly from board-certified physicians , treating them as the absolute ground truth.
>
> ### Question 3:
>
> We sincerely thank the reviewer for this suggestion, as we fully recognize that cross-dataset validation is typically a standard practice for assessing model generalization in the medical domain. However, we respectfully clarify that providing performance metrics on existing spinal datasets (e.g., VerSe, RSNA LumbarDISC) is methodologically infeasible for our specific model due to a fundamental mismatch in task definition and output format. As detailed in our Related Work, prior datasets are predominantly designed for low-level computer vision tasks, requiring outputs such as pixel-level voxel masks for segmentation or binary class labels for detection .
>
> In contrast, SpineGPT is a Large Vision-Language Model (LVLM) architected specifically for high-level clinical reasoning and structured text generation . Our model outputs comprehensive diagnostic reports, treatment plans, and surgical rationales, which cannot be meaningfully evaluated using the geometric metrics (e.g., Intersection-over-Union for segmentation) or simple classification accuracy metrics used by datasets like VerSe or RSNA without fundamentally altering the model's architecture and purpose. Therefore, a direct performance comparison **is unfair****.**
>
> **The limitations of the existing evaluation set are precisely the purpose for which we build SpineBench. SpineBench is currently the only benchmark for evaluating models throughout the entire workflow of spinal diagnosis and treatment, from imaging findings to surgical planning.**

---

### Author Response · Authors · 2025-11-24

**General Response to All Reviewers**

We thank all reviewers for their constructive and encouraging feedback. We appreciate that the reviews recognize **SpineMed-450k**'s unique value in bridging the critical "cognitive gap" between current perception-oriented datasets and the needs of full-spectrum spinal care.

We have made the following major updates to the revised version of the paper accordingly:

* **Section 2.2** now explicitly specifies the "Expert LLM" ensemble (Gemini-2.5-Pro, locally deployed GLM-4.5V, and GPT-5-mini) to ensure full transparency and reproducibility.
* **Section 2.3** adds a dedicated comparison with existing spine datasets (Table 1), highlighting our contribution in moving from unimodal segmentation to multimodal clinical reasoning.
* **Section 5** is restructured as an independent chapter to comprehensively detail the SpineGPT architecture, implementation details, and the rationale behind our Three-Stage Curriculum Learning Framework.
* **Section 6.1** includes new benchmarking results with **Medgemma-27B** and expanded performance analysis to demonstrate the efficiency of our approach.
* **Section 6.3** adds detailed ablation studies empirically proving the specific contributions of general medical priors versus spine-specific data.
* **Appendix B** extends the Related Work to further contextualize the paradigm shift from "Tool AI" to "Collaborator AI".
* **Appendix E**  incorporates comprehensive statistical breakdowns of the SpineBench testing set to empirically validate its rigorous stratification across 14 disease subspecialties and diverse data sources.


All major changes are highlighted in **blue** in the updated PDF. We hope these clarifications and additions address the remaining concerns and strengthen the final version.

---

### Author Response · Authors · 2025-12-01
**Review and Reviewer-Author Discussion Summary**

Dear PCs, SACs, ACs, and Reviewers,

Thank you very much for your valuable contributions to our work. To assist the newly assigned AC and help reduce their workload, we provide below a summary of the key points from the reviews and the reviewer-author discussions.

**Strength.** Overall, we are grateful that the reviewers recognized the clinical value and rigorous construction of our work. Specifically:
* **Pioneering Multimodal Spine Dataset:** The paper contributes SpineMed-450k, the largest clinically-grounded spine dataset, filling a critical gap in specialized medical AI.
    * Three reviewers explicitly highlighted this contribution (**1Ksq**: Strength 2, **8nUb**: Strength 1, **SXG5**: Strength 1 & 2).
* **Rigorous "Clinician-in-the-Loop" Design:** The involvement of 17 board-certified surgeons across data curation and validation was recognized as a key differentiator from standard web-scraped datasets.
    * Reviewers highlighted this medical rigor (**8nUb**: Strength 2, **1Ksq**: Strength 2).
* **Comprehensive Evaluation Framework:** The 10-dimensional evaluation (SpineBench) that assesses communication quality, reasoning, and risk assessment—beyond simple accuracy—was praised.
    * Reviewers noted this better captures real-world utility (**8nUb**: Strength 3, **xkSD**: Soundness 3).
* **Model Efficiency:** The ablation studies effectively demonstrated that a 7B model trained on our data could outperform or rival much larger generalist models on specialized tasks.
    * Reviewers acknowledged this effectiveness (**8nUb**: Strength 4, **1Ksq**: Strength 3).

**Concerns and Our Addressing.** During the discussion period, we actively addressed the reviewers' concerns with new experiments, statistical analyses, and manuscript revisions. Specifically:

**(xkSD: Weakness 4, SXG5: Weakness 2 & 4) Concerns about Benchmark Scale & Robustness.**
* Reviewers questioned the size of the test set (487 QAs, 87 Reports) relative to the training set and potential bias.
* **Our Addressing:** We clarified that SpineBench prioritizes "Gold Standard" purity (expert triple-blind review) over raw volume. In response to **SXG5**'s concern about bias, we added **Appendix E** in the revised paper, providing comprehensive statistical breakdowns. This empirically validates that the test set is rigorously stratified across 14 disease subspecialties (including rare tumors) and diverse data sources. While **SXG5** retained concerns about the absolute size, we believe the stratification analysis confirms the benchmark serves as an unbiased "litmus test" for clinical reasoning.

**(1Ksq: Weakness 2, 8nUb: Weakness 2, SXG5: Weakness 1) Concerns about Model Performance vs. Generalist Models.**
* Reviewers noted that generalist models (e.g., Qwen-72B, GPT-4o) score high on QA tasks, questioning the necessity of a specialized model.
* **Our Addressing:** We highlighted the critical distinction between "Recognition" (QA) and "Reasoning" (Report Generation). We added **Table R4** showing that generalist models collapse on the Report task (Qwen-72B drops from ~82% in QA to **63.8%** in Reports), whereas SpineGPT maintains high performance (**87.24%**). We also added benchmarking results with **Medgemma-27B** in **Section 6.1** to further demonstrate our model's efficiency.

**(8nUb: Weakness 1, 1Ksq: Weakness 1) Concerns about "Level-Aware" Design and Data Necessity.**
* Reviewers asked for the definition of "Level-Aware" and ablation studies to prove the specific value of the spine dataset.
* **Our Addressing:** We defined "Level-Aware" as reasoning anchored to specific vertebral segments. We added **Table R2** and **Table R5** (included in **Section 6.3** of the revision) to empirically prove that models without this design, or trained only on generic medical data, fail to generate valid reports (Score: ~54-65% vs. Our 87%+).
**(xkSD: Weakness 2 & 3, 8nUb: Weakness 3) Concerns about Transparency and Comparisons.**
* Reviewers requested explicit definitions of the "Expert LLM" pipeline and comparisons with existing datasets.
* **Our Addressing:** We updated **Section 2.2** to explicitly specify the ensemble pipeline (Gemini-Pro/Local GLM/GPT-5) and added **Table 1** in **Section 2.3** to benchmark our dataset against existing resources like RSNA and VerSe. We also restructured **Section 5** to detail the baseline model architecture.

**Summary of Revisions.** All major changes, including the new Tables (1, R2, R4, R5) and Appendix E, have been incorporated into the revised PDF and highlighted in **blue**.

Above, we have faithfully summarized all reviewer comments and our corresponding responses. We are deeply grateful to the reviewers, AC, SAC, and PC for their dedicated effort. Their insightful feedback has significantly strengthened the transparency and rigorousness of our work.

Sincerely,

Authors

---

### Meta-Review · Program_Chairs · 2026-01-05

**Summary:**

To address the limitations of AI-assisted diagnosis for spine disorders caused by a lack of specialized, multimodal data, the authors introduce SpineMed-450k and SpineBench. SpineMed-450k claims to be the first large-scale dataset explicitly designed for vertebral-level reasoning, comprising over 450,000 instruction instances derived from diverse sources such as textbooks, guidelines, and approximately 1,000 de-identified hospital cases, all curated through a clinician-in-the-loop pipeline. Complementing this dataset, the authors present SpineBench, a standardized evaluation framework that assesses models on clinically salient axes like pathology assessment, surgical planning, and risk prognosis. The authors’ evals of current Large Vision-Language Models (LVLMs) on SpineBench reveal systematic weaknesses in fine-grained anatomical reasoning. The authors' claim that their fine-tuned model, SpineGPT, demonstrates consistent performance improvements across all tasks.

Reviewers liked the dataset's scale and its unique focus on "level-aware" reasoning, addressing complex diagnostic synthesis. The efficiency of the SpineGPT model, which achieves competitive performance despite its size. However, reviewers noted some weaknesses:(1) small size of the SpineBench test set compared to the training data, (2)lack of clarity regarding the "Expert LLM" pipeline and the "level-aware" design, (3) performance gap between the specialized SpineGPT and generalist models not sufficiently large (4) small test data. I think this is a good paper in the right direction despite some shortcomings of small test data and useful for medical AI. I recommend acceptance if the ethics review is passed.

**This paper is being conditionally accepted provided the authors address the following in the camera-ready**:
[Ethics concerns] The authors should disclose in detail their data sources, providing as much detail as possible without releasing PII.
** Conditions for acceptance have been satisfied**.

**Reviewer Concerns:**

The authors addressed the ambiguity surrounding the "Expert LLM" by explicitly defining the ensemble pipeline in their revision I think. They also countered the skepticism regarding the "level-aware" design and dataset necessity with new ablation studies. However, the concern regarding the small scale of the test set remains outstanding for at least one reviewer

**Reviewer Scores:**

- Reviewer xkSD (Current: 4): Likely 5/6. The authors provided the requested dataset comparisons, a dedicated baseline section, and clarified the "Expert LLM" definition.
- Reviewer 1Ksq (Current: 8): Likely remain 8
- Reviewer 8nUb (Current: 4): Likely 5/6. The definitions for "level-aware" and "Expert LLM" were provided, and new ablation studies showed  specialized model's efficiency over generalist models.
Reviewer SXG5 (Current: 4): Likely to remain 4. this reviewer explicitly noted that "concerns remain" regarding the test dataset setting

---

> ### Public Comment · ~Wenhui_Dong1 · 2026-03-02
> **Ethics and Data Source Disclosures**
>
> In the camera-ready version, we have provided a comprehensive disclosure of all data sources in Appendix A.1. We detail the curation and processing pipeline for each data subset, providing maximum transparency while rigorously ensuring that no Personally Identifiable Information (PII) is released.

---

### Decision · Program_Chairs · 2026-01-26

Accept (Poster)